# A *Cautionary Tale* on Integrating Studies with Disparate Outcome Measures for Causal Inference

**Harsh Parikh**
Yale University
`harsh.parikh@yale.edu`

**Trang Quynh Nguyen**
Johns Hopkins University
`trang.nguyen@jhu.edu`

**Elizabeth A. Stuart**
Johns Hopkins University
`estuart@jhu.edu`

**Kara Rudolph**
Columbia University
`kr2854@cumc.columbia.edu`

**Caleb Miles**
Columbia University
`cm3825@cumc.columbia.edu`

## Abstract

Data integration approaches are increasingly used to enhance the efficiency and generalizability of studies. However, a key limitation of these methods is the assumption that outcome measures are identical across datasets – an assumption that often does not hold in practice. Consider the following opioid use disorder (OUD) studies: the XBOT trial and the POAT study, both evaluating the effect of medications for OUD on withdrawal symptom severity (not the primary outcome of either trial). While XBOT measures withdrawal severity using the subjective opiate withdrawal scale, POAT uses the clinical opiate withdrawal scale. We analyze this realistic yet challenging setting where outcome measures differ across studies and where neither study records both types of outcomes. Our paper studies whether and when integrating studies with disparate outcome measures leads to efficiency gains. We introduce three sets of assumptions – with varying degrees of strength – linking both outcome measures. Our theoretical and empirical results highlight a cautionary tale: integration can improve asymptotic efficiency only under the strongest assumption linking the outcomes. However, misspecification of this assumption leads to bias. In contrast, a milder assumption may yield finite-sample efficiency gains, yet these benefits diminish as sample size increases. We illustrate these trade-offs via a case study integrating the XBOT and POAT datasets to estimate the comparative effect of two medications for opioid use disorder on withdrawal symptoms. By systematically varying the assumptions linking the SOW and COW scales, we show potential efficiency gains and the risks of bias. Our findings emphasize the need for careful assumption selection when fusing datasets with differing outcome measures, offering guidance for researchers navigating this common challenge in modern data integration.

## 1 Introduction

Robust decision-making increasingly depends on integrating information from diverse sources – a practice commonly referred to as *data integration*. By harnessing complementary datasets, researchers can improve the accuracy, generalizability, and efficiency of statistical inference (Bareinboim and Pearl, 2016). In the realm of causal inference, data integration has emerged as a central focus, recently cited among the top ten priorities for advancing the field (Mitra et al., 2022). This surge of interest reflects its wide-ranging utility: from generalizing or transporting evidence (Degtiar and Rose, 2023; Parikh et al., 2024; Huang and Parikh, 2024), to heterogeneous causal effect estimation (Brantner

39th Conference on Neural Information Processing Systems (NeurIPS 2025).

et al., 2023), boosting statistical efficiency (Rosenman et al., 2023), and mitigating bias (Kallus et al., 2018).

However, in many real-world scenarios, various data sources may capture outcomes that, while related, are not identical to those measured in the trial. For example, in studies on medications for opioid use disorder (MOUD), the intensity of withdrawal symptoms can be measured using two different scales: the Clinical Opiate Withdrawal Scale (COWS) and the Subjective Opiate Withdrawal Scale (SOWS) (Wesson and Ling, 2003; Handelsman et al., 1987). In the XBOT trial that compared the effectiveness of injection naltrexone to sublingual buprenorphine in terms of reducing risks of returning to regular opioid use, withdrawal symptoms were measured using SOWS (Lee et al., 2018). However, the POATS study, which compared the effectiveness of adding counseling to sublingual buprenorphine treatment, used COWS to measure the strength of withdrawal symptoms (Weiss et al., 2010). Despite the differences in outcome measures, researchers might wish to leverage the POATS study to improve the precision of treatment effect estimates in the XBOT trial (or vice versa). This raises an important question: *when can integration of primary study and auxiliary data with disparate outcome measures yield efficiency gains for causal effect estimates if neither study has observation of both outcome measures on the same group of individuals?*

**Contributions.**   Our paper addresses this question by examining scenarios in which neither the trial nor the auxiliary data records both outcome measures on the same set of individuals.

- We formulate a principal assumption that connects the primary outcome in the trial with the auxiliary outcome in external data, offering a conceptual "license" to borrow strength from auxiliary sources. We present three versions of this assumption – ranging from strong to weak – thereby providing a flexible framework that reflects varying degrees of identifiability.

- We characterize the conditions under which integrating studies can improve semiparametric efficiency as well as finite sample gains. We show that asymptotic gains are only possible under the strongest assumptions (albeit at a risk of some bias). However, under milder (and perhaps more realistic) conditions, finite-sample improvements may be realized, although these benefits diminish as sample sizes grow.

- We illustrate these insights through simulation studies and a real-world case study from the MOUD trial. Our findings underscore both the promise and the limitations of using auxiliary data with non-overlapping outcomes. Importantly, we provide practical guidance for researchers aiming to navigate these tradeoffs in applied causal inference settings.

In a nutshell, this paper presents a cautionary framework for data integration in the presence of disparate outcomes, showing that while such integration may yield marginal gains under ideal conditions, it carries a significant risk of bias when assumptions are violated – as illustrated by our case study. To the best of our knowledge, we present the first formal quantification of this tradeoff, emphasizing the need for scrutiny before applying such methods in practice.

The paper is organized as follows. Section 3 introduces the notation, setup, and standard assumptions. Section 4 presents the key structural assumption linking primary and auxiliary outcomes, along with three scenarios that reflect varying degrees of prior knowledge about this relationship. Sections 4.2– 4.4 contains our main theoretical contributions: semiparametric efficiency bounds under each scenario, as well as worst-case bounds on finite-sample estimation errors. In Section 5, we apply these methods to estimate the causal effect of medications for opioid use disorder (MOUD) on withdrawal severity, using SOWS (from the XBOT trial) and COWS (from the POAT study). Section 6 concludes with a summary of key findings, limitations, and directions for future research. Appendix A presents simulation results evaluating estimator performance across varying sample sizes and dimensions. Additional theoretical discussion and proofs are provided in Appendices B, C, and D.

## 2   Relevant Literature

We briefly review four bodies of literature related to our work: (i) data integration in causal inference, (ii) meta-analysis, (iii) data harmonization, and (iv) surrogate outcomes.

**Data Integration for Causal Inference.**   Data integration has emerged as a central focus in causal inference, recently cited among the top priorities for advancing the field (Mitra et al., 2022). It

supports a wide range of goals, including generalizing evidence across populations (Degtiar and Rose, 2023; Pearl, 2015; Parikh et al., 2024; Huang and Parikh, 2024), estimating heterogeneous effects (Brantner et al., 2023), boosting efficiency (Li and Luedtke, 2023), and mitigating bias (Kallus et al., 2018; Parikh et al., 2023b). Recent methods improve efficiency by combining auxiliary datasets while controlling bias, such as James-Stein shrinkage (Rosenman et al., 2023), semiparametric estimators (Yang et al., 2020), bias correction (Kallus et al., 2018; Yang and Ding, 2020), and Bayesian borrowing (Lin et al., 2024). However, these approaches typically assume consistent measures – including outcomes – across datasets.

**Meta-Analysis and Evidence Synthesis.** When outcomes differ, naïve pooling can induce substantial bias (Van Cleave et al., 2011). Early evidence synthesis methods, such as standardizing outcomes (Murad et al., 2019; Deeks et al., 2019), rely on strong equivalence assumptions. Traditional meta-analyses, as in (Deeks et al., 2019), uses heuristics like dichotomization or normalization, assuming commensurability across studies (Murad et al., 2019). More sophisticated approaches jointly model multiple outcomes, using multivariate Bayesian methods (Bujkiewicz et al., 2016) or multi-task learning analogs (Zhang and Yang, 2018). These frameworks exploit known outcome dependencies or co-measurement of outcomes to synthesize information while allowing outcome-specific variation.

**Data Harmonization.** Data harmonization methods are a set of tools that aim to equate measures across data sources to facilitate data integration. These methods typically align heterogeneous outcomes through co-calibration (Nance et al., 2017) or latent constructs (Snavely et al., 2014). Bridge studies, where multiple outcomes are measured on the same set of individuals, can estimate mappings between outcome measures, while latent variable models treat observed outcomes as noisy indicators of a shared construct. These approaches typically require both outcome measurements for the same individual and introduce additional modeling assumptions.

**Leveraging Surrogate Outcomes.** Another relevant literature is on data integration methods leveraging studies with surrogate outcomes. For instance, Athey et al. (2019) and Ghassami et al. (2022) combine experimental data with short-term outcome measures with an observational study where long-term outcome is measured to yield a consistent estimate of the long-term treatment effect. Surrogate indices that aggregate multiple proxies can substantially improve efficiency (Ghassami et al., 2022), but rely on strong structural assumptions about the proxy–outcome relationship. Existing methods generally require at least one dataset with measurement of primary and surrogate outcomes on the same set of individuals – an assumption often violated in practice and one that motivates our work.

## 3   Preliminaries

**Setup and Notations.** We consider two studies: a *primary study* ($S = 0$) and an *auxiliary study* ($S = 1$). The primary study observes the outcome of interest $Y$, while the auxiliary study observes a related but distinct outcome $W$. Crucially, $Y$ and $W$ are never observed for the same individual. In both studies, we observe treatment $T \in \{0, 1\}$ and covariates $X$. Let $Y(t)$ and $W(t)$ denote the potential outcomes under treatment $T = t$. To unify notation, define the observed outcome as $V := (1 - S)Y + SW$, and the observed data as $O := (X, S, T, V)$. We let $\mathcal{S}_n = \{O_1, \ldots, O_n\}$ denote a sample of $n$ units, with $n_0$ and $n_1$ representing the number of units in the primary and auxiliary studies, respectively.

For any (random) function $f$, let $\mathbb{E}[f(A)]$ denote the expectation, $\mathcal{P}_n(f(A)) = \frac{1}{n} \sum_{i=1}^{n} f(A_i)$ the empirical average, and $\mathcal{P}(f(A)) = \int f(a) \, dP(a)$ the population average treating $f$ as fixed. Note that $\mathbb{E}[f(A)]$ integrates over randomness in both $A$ and $f$, while $\mathcal{P}(f(A))$ treats $f$ as fixed. We also define the $L^q(P)$ norm as $\|f\|_q = \left( \int |f(o)|^q \, dP(o) \right)^{1/q}$. Futher, for compactness, we write $\mu_A(B = b) := \mathbb{E}[A \mid B = b]$ to denote the conditional expectation of $A$ given $B = b$, and $\nu_A^t(B = b) := \mathbb{E}[A(t) \mid B = b]$ for the conditional mean of the potential outcome $A(t)$.

Our goal is to estimate the *conditional average treatment effect (CATE)*: $\tau_0(x) := \nu_Y^1(X = x) - \nu_Y^0(X = x)$, and the *average treatment effect (ATE)*: $\tau_0 := \nu_Y^1 - \nu_Y^0$, both defined with respect to the primary outcome $Y$.

**Assumptions & Identification** We make the following assumptions:

A.1. *(S-ignorability)* $\forall x, \quad Y(t), W(t) \perp S \mid X = x$.

A.2. *(Treatment Positivity)* $\epsilon < P(T = t \mid X, S = 0) < 1 - \epsilon$, for all $t \in \{0, 1\}$.

A.3. *(Sampling Positivity)* $\epsilon < P(S = 0 \mid X) < 1 - \epsilon$.

A.4. *(Conditional Ignorability)* $\forall x, s, \quad Y(t), W(t) \perp T \mid X = x, S = s$.

We assume the following structural models for the potential outcomes:

$$Y(t) = \theta(X)t + g(X) + \gamma, \quad \gamma \sim \mathcal{N}(0, \sigma_Y^2), \tag{1}$$

$$W(t) = \phi(X)t + f(X) + \delta, \quad \delta \sim \mathcal{N}(0, \sigma_W^2), \tag{2}$$

where $\theta(X)$ and $\phi(X)$ are the treatment effect functions for $Y$ and $W$, respectively. These formulation is commonly used in the causal inference literature (Robinson, 1988; Chernozhukov et al., 2018; Hahn et al., 2020; Rudolph et al., 2025). From Equation (1), it follows that the CATE in the primary population is: $\tau_0(x) = \mathbb{E}[Y(1) - Y(0) \mid X = x] = \theta(x)$. By Assumptions A.2. and A.4., the potential outcome means $\nu_Y^t(X = x)$ are *identified* by observed data as: $\nu_Y^t(X = x) = \mu_Y(X = x, T = t)$. Hence, the CATE is identified as: $\tau_0(x) = \mu_Y(X = x, T = 1) - \mu_Y(X = x, T = 0)$.

**Influence Function.** Let $\eta$ denote the collection of nuisance parameters, specifically $\eta = \{\mu_V(X, S), \mu_T(X, S), \mu_S(X)\}$. Let $\theta_0$ be the true parameter of interest, and $\eta_0$ the true nuisance parameters governing the data-generating process. For any regular, consistent, and asymptotically linear estimator $\hat{\theta}$ of $\theta_0$, there exists a function $\psi$ – called the *influence function* – such that we decompose the estimation error $\hat{\theta} - \theta_0$ using the von Mises expansion as:

$$\hat{\theta} - \theta_0 = \underbrace{(\mathcal{P}_n - \mathcal{P})\psi(O; \theta_0, \eta_0)}_{M_1} - \underbrace{\mathcal{P}\left[\psi(O; \theta_0, \hat{\eta}) - \psi(O; \theta_0, \eta_0)\right]}_{M_2(\hat{\eta})} + M_3(\hat{\eta}) \quad [1]$$

where $\mathcal{P}[\psi(O; \theta_0, \eta_0)] = 0$ (Tsiatis, 2006; Kennedy, 2016). Here, (i) the *first term*, $M_1$, represents sampling variability resulting in asymptotic variance – capturing the first-order behavior of $\hat{\theta}$ and reflects its asymptotic linearity (Ichimura and Newey, 2022; Kennedy, 2016); (ii) the *second term*, $M_2(\hat{\eta})$, captures bias due to finite sample estimation of nuisance functions; (iii) the *third term*, $M_3(\hat{\eta})$, accounts for remaining higher-order approximation error that converges to 0 in probability at rate faster than $\sqrt{n}$.

By the Central Limit Theorem and the Slutsky theorem, the estimator is asymptotically normal (provided Donsker condition holds or sample splitting is used):

$$\sqrt{n}(\hat{\theta} - \theta_0) \rightsquigarrow \mathcal{N}\left(0, \mathbb{E}[\psi(O; \theta_0, \eta_0)\psi(O; \theta_0, \eta_0)^T]\right).$$

The asymptotic variance of $\hat{\theta}$ is thus determined by the variance of the influence function and can be consistently estimated via the empirical variance of the estimated influence function, $\hat{\psi}$.

**Efficient Influence Function.** The influence function depends on the values of $\theta$ and $\eta$, although we suppress this dependency for notational convenience. Among all influence functions corresponding to regular, asymptotically linear estimators of $\theta_0$, the *efficient influence function* (EIF), denoted $\psi^*$, achieves the smallest possible asymptotic variance. This minimal variance – known as the *semiparametric efficiency bound* – is given by: $\mathbb{E}[\psi^*(O; \theta_0, \eta_0)\psi^*(O; \theta_0, \eta_0)^T]$, and represents the best achievable precision for unbiased estimation (Tsiatis, 2006; Newey, 1990).

**Procedure to Derive EIF.** Consider the log-likelihood $\mathcal{L}(O; \theta, \eta)$ of observed data with parameter of interest $\theta$ and nuisance parameters $\eta$, maximized at $(\theta_0, \eta_0)$. We define the score functions with respect to $\theta$ and $\eta$ as $R_\theta(O; \theta_0, \eta_0) = \frac{\partial \mathcal{L}}{\partial \theta}\big|_{\theta_0, \eta_0}$ and $R_\eta(O; \theta_0, \eta_0) = \frac{\partial \mathcal{L}}{\partial \eta}\big|_{\theta_0, \eta_0}$, respectively. $R_\theta$ reflects the sensitivity of the likelihood to $\theta$. However, it may also be sensitive to $\eta$. Projecting it orthogonally to the space spanned by $R_\eta$ isolates the component of information unique to $\theta$. The efficient score function is the residual of $R_\theta$ after projecting out components in the linear span of $R_\eta$:

$$R^*(O; \theta_0, \eta_0) = R_\theta - \Pi(R_\theta | \Lambda_\eta),$$

---

[1] Here, $\mathcal{P}_n(\psi)$ denotes the empirical average and $\mathcal{P}(\psi)$ the population expectation. The notation $(\mathcal{P}_n - \mathcal{P})(\psi)$ is shorthand for $\mathcal{P}_n(\psi) - \mathcal{P}(\psi)$, as commonly used in semiparametric theory.

where $\Pi\left(R_\theta, |, \Lambda_\eta\right) = \mathbb{E}\left[R_\theta R_\eta^T\right]\left\{\mathbb{E}\left[R_\eta R_\eta^T\right]\right\}^{-1} R_\eta$ and arguments $(O; \theta_0, \eta_0)$ are suppressed for brevity. The efficient influence function is given by $\psi^* = \left\{\mathbb{E}\left[R^*(R^*)^T\right]\right\}^{-1} R^*$. This influence function achieves the semiparametric efficiency bound and serves as the optimal estimating function for $\theta$ under the given model. For further discussion and derivation, we refer readers to Tsiatis (2006).

# 4 Data Integration with Disparate Outcome Measures

To leverage auxiliary data for estimating treatment effects on the primary outcome $Y$, we must establish a relationship between $Y$ and the auxiliary outcome $W$. We posit the following structural assumption that provides a foundation – or "license" – for incorporating $W$ into the analysis:

A.5. *(Outcome Link Assumption)* For all $x$ and $t$, there exist functions $\alpha$ and $\beta$ of pre-treatment covariates such that $\nu_Y^t(x) = \alpha(x)\nu_W^t(x) + \beta(x)$

*Remark* 1 (*On Assumption A.5.*). The assumption allows for flexible and heterogeneous relationships between primary and auxiliary outcomes across units with different values of $X$. However, this assumption also imposes structural restrictions on the relationship: the primary outcome is a partially linear function of the auxiliary outcome $W$, with the scaling factor $\alpha(X)$ and shift $\beta(X)$ modulated by pre-treatment covariates $X$.

This assumption is plausible in settings where $W$ serves as a meaningful proxy for $Y$. For instance, in biomedical studies, $W$ might represent a surrogate endpoint (e.g., a biomarker) that reflects the underlying disease progression captured by $Y$ (Weir and Walley, 2006). In such cases, prior studies or mechanistic understanding can inform how changes in $W$ relate to changes in $Y$.

## 4.1 Assumption Sets on $\alpha(X)$ and $\beta(X)$

To explore the range of identifiability and efficiency in leveraging auxiliary data, we consider three increasingly weaker assumptions about prior knowledge about $\alpha(X)$ and $\beta(X)$:

A.5(a) *Fully Known Link*: Both $\alpha(X)$ and $\beta(X)$ are known from prior domain knowledge.

A.5(b) *Partially Known Link*: Only $\beta(X)$ is known; $\alpha(X)$ is unknown.

A.5(c) *Unknown Link*: Neither $\alpha(X)$ nor $\beta(X)$ are known.

*Assumption A.5(a)* represents the strongest assumption, and is most tenable in domains with well-characterized mechanistic knowledge – such as certain areas of biology, pharmacology, or engineering – where $\alpha(X)$ and $\beta(X)$ are grounded in empirical studies or physical theory (Puniya et al., 2018; Parikh et al., 2023a). In such contexts, auxiliary outcomes can be confidently incorporated using known mappings to the primary outcome. *Assumption A.5(b)* relaxes this requirement by assuming only the baseline shift $\beta(X)$ is known. This is common in applications where historical data or expert knowledge informs baseline trends, but the strength of association (i.e., scaling) between $Y$ and $W$ varies across populations or settings. Such partial knowledge arises frequently in social sciences or public health (Handelsman et al., 1987). *Assumption A.5(c))* is the most general and aligns with many real-world scenarios where no prior information is available about the relationship between $Y$ and $W$. This assumption allows maximum flexibility, but also introduces the greatest challenge in using an auxiliary study.

These assumptions represent a spectrum of tradeoffs between realism and statistical precision. Stronger assumptions enable tighter and more efficient estimation but rely more heavily on prior knowledge. We make this tradeoff explicit in Sections 4.2, 4.3 and 4.4. Ultimately, the appropriate assumption set depends on the context and credibility of available domain knowledge.

**Function Class Complexity Assumption.** We make additional assumptions about the complexity of functions in A.5.:

A.6. There exists positive constants $\varepsilon > 0$ such that $\mathcal{A}$ and $\mathcal{B}$ satisfies the covering number bound: $\log N(\varepsilon, \mathcal{A}, \|\cdot\|) = O(\varepsilon^{-\omega_\alpha})$, and $\log N(\varepsilon, \mathcal{B}, \|\cdot\|) = O(\varepsilon^{-\omega_\beta})$. Further, we assume that the function class for $\mu_Y$ and $\mu_W - \mathcal{M}$ – satisfies covering number bounds: $\log N(\varepsilon, \mathcal{M}, \|\cdot\|) = O(\varepsilon^{-\omega})$, with $\omega_\alpha + \omega_\beta \leq \omega$.

This assumption imposes regularity conditions on the function classes involved in the decomposition of $\mu_Y(X)$ into $\alpha(X)$ and $\beta(X)$. Specifically, it ensures that the combined complexity of $\alpha$ and $\beta$, measured via covering number bounds, does not exceed that of $\mu_Y$. This is a mild and natural requirement: if the auxiliary outcome $W$ is informative about $Y$, then the residual mapping captured by $\alpha(X)$ is expected to be simpler than modeling $\mu_Y(X)$ directly. In this sense, Assumption A.6. reflects a form of functional regularization, where using a predictive surrogate reduces the effective complexity of the learning task.

## 4.2 Semiparametric Efficiency Bounds

Now, we derive the efficient bounds under each of the following three assumptions A.5(a), A.5(b), and A.5(c), and investigate if and when data integration yields semiparametric efficiency gains. Throughout this section, we assume that assumptions A.1. to A.4. hold.

Recall $\psi^*(O; \theta_0, \eta_0) = \left\{ \mathbb{E}[R^*(O; \theta_0, \eta_0) R^*(O; \theta_0, \eta_0)^T] \right\}^{-1} R^*(O; \theta_0, \eta_0)$, and the semiparametrically efficient asymptotic variance (i.e., efficiency bound) is equal to $\left\{ \mathbb{E}[R^*(O; \theta_0, \eta_0) R^*(O; \theta_0, \eta_0)^T] \right\}^{-1}$, we only present the efficient score function $R^*$ instead of the EIF $\psi^*$. However, note that deriving the EIF from $R^*$ is straightforward in our context. In our case, $P(O = o; \theta, \eta) = P(X = x)P(S = s \mid X = x)P(T = t \mid X = x, S = s)P(V = v \mid T = t, X = x, S = s)$ and $\mathcal{L}(O; \theta, \eta) = \log P(O = o; \theta, \eta)$.

First, we derive the efficiency bound for the semiparametrically efficient estimator that only uses the primary study. We use this result as a base case to compare the efficiency bounds for the data integration-based estimators. This efficiency bound is akin to the one derived in (Robinson, 1988).

**Theorem 1 (Efficiency bound using only primary data).** *Under assumptions A.1.–A.4., the efficient score function using only the primary study ($S = 0$) is $R_0^*(O; \theta_0, \eta_0) = (1 - S) \cdot \Delta_0$. The corresponding asymptotic variance is $\mathbb{V}_0^\theta(X) = \left( \mathbb{E}\left[ \Delta_0^2 \mid S = 0, X \right] p(S = 0 \mid X) \right)^{-1}$, where $\Delta_0 = \left( (V - \mu_Y(X, 0) - \theta(X)(T - \mu_T(X, 0))) \cdot \frac{T - \mu_T(X, 0)}{\sigma_Y^2} \right)$.*

Now, we derive the efficiency bound that leverages auxiliary data under A.5(a).

**Theorem 2 (Efficiency bound under known $\alpha(X)$ and $\beta(X)$).** *Under assumptions A.1.–A.4. and A.5(a), the efficient score function is:*

$$R_a^*(O; \theta_0, \eta_0) = S \cdot \Delta_1 + (1 - S) \cdot \Delta_0,$$

*where*

$$\Delta_1 = (\alpha(X)(V - \mu_W(X, 1)) - \theta(X)(T - \mu_T(X, 1))) \cdot \frac{T - \mu_T(X, 1)}{\alpha^2(X)\sigma_W^2}.$$

*The asymptotic variance is:*

$$\mathbb{V}_a^\theta(X) = \left( \mathbb{E}\left[ \Delta_0^2 \mid S = 0, X \right] p(S = 0 \mid X) + \mathbb{E}\left[ \Delta_1^2 \mid S = 1, X \right] p(S = 1 \mid X) \right)^{-1}.$$

*Corollary* 1. Integrating primary and auxiliary data under assumption A.5(a) yields efficiency gain i.e. $\mathbb{V}_a^\theta(X) \leq \mathbb{V}_0^\theta(X)$.

Next, we derive the efficiency score and the efficiency bound for a case when A.5(b) holds.

**Theorem 3 (Efficiency bound under known $\beta(X)$ only).** *Under assumptions A.1.–A.4. and A.5(b), the efficient score function for $\theta(X)$ and $\alpha(X)$ is:*

$$R_b^*(O; \theta_0, \alpha_0, \eta_0) = \begin{pmatrix} S \cdot \Delta_1 + (1 - S) \cdot \Delta_0 \\ S \cdot \left( \frac{\theta(X)(T - \mu_T(X, 1)) - \alpha(X)(V - \mu_W(X, 1))}{\alpha^2(X)\sigma_W^2} \cdot \frac{\theta(X)(T - \mu_T(X, 1))}{\alpha(X)} \right) \end{pmatrix}.$$

*The corresponding asymptotic variance-covariance matrix is:*

$$\boldsymbol{\Sigma}_b(X) := \begin{pmatrix} \mathbb{V}_b^\theta(X) & \text{Cov}_b^{\theta, \alpha}(X) \\ \text{Cov}_b^{\theta, \alpha}(X) & \mathbb{V}_b^\alpha(X) \end{pmatrix}, \text{ with } \mathbb{V}_b^\theta(X) = \left( \mathbb{E}[\Delta_0^2 \mid S = 0, X] P(S = 0 \mid X) \right)^{-1}.$$

*Corollary* 2. The asymptotic variance of the efficient estimator of $\theta(X)$ under assumption A.5(b) is equal to that under using primary data only: $\mathbb{V}_b^\theta(X) = \mathbb{V}_0^\theta(X)$. Thus, when $\alpha(X)$ is unknown, incorporating auxiliary data provides no efficiency gain.

**Theorem 4** (**Efficiency bound under unknown** $\alpha(X)$ **and** $\beta(X)$). *Under assumptions A.1.–A.4. and A.5(c), the efficient score function is identical to that in Theorem 3:* $R_c^*(O; \theta_0, \alpha_0, \eta_0) = R_b^*(O; \theta_0, \alpha_0, \eta_0)$. *Therefore, the asymptotic variance for estimating* $\theta(X)$ *remains:* $\mathbb{V}_c^\theta(X) = \mathbb{V}_b^\theta(X) = \mathbb{V}_0^\theta(X)$.

*Corollary* 3. If both $\alpha(X)$ and $\beta(X)$ are unknown, there are no efficiency gains from using auxiliary data compared to using only primary data.

The proofs and results in Theorems 1 to 4 are a direct consequence of following the procedure to derive EIF described in Section 3 and are provided in Appendix B.

### 4.3 ATE Estimation under A.5(a)

Now, we present the ATE estimation under assumption A.5(a). We use the efficient score $R_a^*$ to guide the estimation of $\theta_0$ using the property that $\mathbb{E}[R_a^*(O; \theta_0, \eta_0)] = 0$. Recall, that $R_a^*(O; \theta_0, \eta_0) = S\Delta_1 + (1 - S)\Delta_0$. Assuming an unbiased and consistent estimate of the nuisance parameter $\hat{\eta}$, a solution to $\frac{1}{n_0} \sum_i R_a^*(O_i; \theta, \hat{\eta})$ – denoted by $\hat{\theta}_a$ – is an unbiased and consistent estimate of $\theta_0$. Let $r_A(B) := A - \mu_A(B)$ denote the residual of random variable $A$ after regressing $A$ on $B$. Then, the estimator $\hat{\theta}_a$ is given by:

$$\hat{\theta}_a = \frac{\sum_i \left( (1 - S_i)\frac{\hat{r}_Y(X_i, 0)\hat{r}_T(X_i, 0)}{\hat{\sigma}_Y^2} + S_i\frac{\hat{r}_W(X_i, 1)\hat{r}_T(X_i, 1)}{\alpha(X_i)\hat{\sigma}_W^2} \right)}{\sum_i \left( (1 - S_i)\frac{\hat{r}_T^2(X_i, 0)}{\hat{\sigma}_Y^2} + S_i\frac{\hat{r}_T^2(X_i, 1)}{\alpha^2(X_i)\hat{\sigma}_W^2} \right)}$$

**Misspecification Bias under A.5(a):** We showed the efficiency bound under three varied assumptions and our results highlighted that efficiency gain is only feasible under the strongest assumption. Now, we investigate the cost of making the wrong assumption i.e. what happens if we assume A.5(a) but $\alpha$ is misspecified. Let $\alpha^\star$ denote the true $\alpha$ and $\alpha_{mis}$ denote a misspecified $\alpha$.

**Theorem 5** (Misspecification Bias). *Under assumptions A.1. – A.4. and a misspecified A.5(a), the estimator,* $\hat{\theta}_a$, *is biased where the bias is equal to* $\mathbb{E}\left[B(X) \mid S = 1\right]$, *where*

$$B(X) := \mathbb{E}\left[ \left( \frac{(\alpha_{mis}(X) - \alpha^\star(X))}{\alpha^\star(X)} \right) \theta(X) \mid S = 1, X \right]$$

### 4.4 Estimation under A.5(b) and A.5(c): Finite-Sample Gains

In cases when $\alpha$ is unknown (i.e. A.5(b) and A.5(c)), it is not feasible to yield efficiency gains by leveraging auxiliary data. However, consider the estimator for ATE only using primary data

$$\hat{\theta}_0 = \frac{\sum_i (1 - S_i)[(\hat{r}_Y(X_i, 0))(\hat{r}_T(X_i, 0))]}{\sum_i (1 - S_i)[(\hat{r}_T(X_i, 0))^2]}.$$

This estimator can be modified, under A.5., to use the auxiliary data to potentially have finite sample benefits. One natural approach to leveraging auxiliary data is the following two-stage estimator: in the first stage, we estimate the auxiliary regression $\mu_W(X, 1) = \mathbb{E}[W|X, S = 1]$ using the auxiliary data and we then use this estimated function to predict $\hat{\mu}_W(X, 0)$ for units in the primary data. In the second stage, we estimate $\mu_Y$ as: $\hat{\mu}_{Y,b}(X, 0) = \hat{\alpha}(X)\hat{\mu}_W(X, 0) + \hat{\beta}(X)$ where $\hat{\alpha}, \hat{\beta} \in \left[\arg\min_{\alpha, \beta \in \mathcal{A}, \mathcal{B}} \frac{1}{n_0} \sum_i (1 - S_i)\left(Y_i - \alpha(X_i)\hat{\mu}_W(X_i, 0) - \beta(X_i)\right)^2\right]$. The resulting fitted function $\hat{\mu}_{Y,b}(X, 0)$ combines both sources of information and provides a data-adaptive estimator of the conditional mean outcome in the primary population. This approach is akin to adjusting for the prognostic or benefit score along with the vector of covariates (Liao et al., 2025). Thus, the resulting estimator leveraging the auxiliary data is given as:

$$\hat{\theta}_b = \frac{\sum_i (1 - S_i)[(\hat{r}_{Y,b}(X_i, 0))(\hat{r}_T(X_i, 0))]}{\sum_i (1 - S_i)[(\hat{r}_T(X_i, 0))^2]}.$$

**Quantifying Finite-Sample Risk.** Asymptotically, if nuisance estimators $\hat{\eta}$ belong to a Donsker class or are fit using sample splitting, the $M_2$ and $M_3$ vanishes asymptotically at a rate

faster than $\sqrt{n}$. However, in finite samples, they contribute non-negligibly to estimation error. We focus on $M_2(\hat{\eta}) = \mathcal{P}(\psi(O; \theta_0, \hat{\eta}) - \psi(O; \theta_0, \eta_0))$, which depends on the accuracy of nuisance function estimates. For cross-fitted estimators, we have: $|M_2(\hat{\eta})| = o_p\left(n^{-1/2}\left(\|\mu_Y - \hat{\mu}_Y\| \cdot \|\mu_T - \hat{\mu}_T\| + \theta_0 \|\mu_T - \hat{\mu}_T\|^2\right)\right)$. Since the only difference between $\hat{\theta}_0$ and $\hat{\theta}_b$ lies in the choice of outcome regression, smaller $\|\mu_Y - \hat{\mu}_Y\|$ directly translates into precise estimates.

**Theorem 6** (**Error bound for $\hat{\mu}_Y$**). *Given assumptions A.1.–A.4., A.5(b), and A.6., the empirical errors for $\hat{\mu}_{Y,0}$ and the two-stage estimator $\hat{\mu}_{Y,b}$ are*

$$\|\hat{\mu}_{Y,0} - \mu_Y\| = o_p\left(n_0^{-\frac{1}{2+\omega}}\right), \text{ and } \|\hat{\mu}_{Y,b} - \mu_Y\| = o_p\left(n_0^{-\frac{1}{2+\omega}}\left(n_0^{\frac{1}{2+\omega} - \frac{1}{2+\omega_\alpha}} + (n_1/n_0)^{-\frac{1}{2+\omega}}\right)\right).$$

**Characterizing Finite Sample Gains.** Theorem 6 demonstrates that one may not even achieve finite sample gains when leveraging auxiliary data. The two-stage estimator $\hat{\mu}_{Y,b}$ can outperform the direct regression estimator $\hat{\mu}_{Y,0}$ only when certain structural and sample size conditions are met. *Qualitatively,* gains arise when leveraging the auxiliary data allow for decomposition of $\mu_Y(X)$ into less complex functions. Additionally, leveraging auxiliary data helps only if $\mu_W(X)$ can be estimated accurately – that is, when the auxiliary sample size $n_1$ is sufficiently large relative to the primary sample size $n_0$. Importantly, when the auxiliary outcome $W$ is highly predictive of the primary outcome $Y$ – that is, when $\text{Cov}(Y, W \mid X)$ is large – the function $\mu_Y(X)$ can be well-approximated by $\mu_W(X)$ then the function $\alpha(X)$ captures only residual structure and tends to be significantly simpler than $\mu_Y(X)$ itself, implying that the entropy exponent $\omega_\alpha$ is relatively much smaller than $\omega$.

*Quantitatively,* finite-sample improvement occurs when $n_0^{\frac{1}{2+\omega} - \frac{1}{2+\omega_\alpha}} + (n_0/n_1)^{\frac{1}{2+\omega}} < 1$. The first term captures the gain from replacing the full function class $\mathcal{M}_Y$ with a lower-complexity class $\mathcal{A}$, and the second term reflects the accuracy of estimating $\mu_W(X)$ from the auxiliary data. Gains are most pronounced when $\omega_\alpha \ll \omega$ (i.e., $\alpha(X)$ is much simpler than $\mu_Y(X)$) and when $n_1 \gg n_0$ (i.e., we have ample auxiliary data). Our characterization formally supports the intuition that structural assumptions and additional data may result in finite sample gains.

In summary, finite-sample efficiency gains from incorporating auxiliary data arise when the function $\alpha(X)$, capturing the dependence between Y and W, is simpler to estimate than $\mu_Y(X)$. In such cases, one may first estimate $\mu_W(X)$ using auxiliary data, and then use the combined data to estimate $\alpha(X)$. However, the existence and extent of these gains depend on the relative complexity of the function classes and the sample sizes involved. As $n_0$ increases, the benefit of this two-stage strategy diminishes, and asymptotically, both the direct and the auxiliary-based estimators converge to the same efficiency.

# 5 Medication for Opioid Use Disorder and Withdrawal Symptoms

We apply our framework to compare the effectiveness of extended-release naltrexone (XR-NTX) and buprenorphine-naloxone (BUP-NX) in reducing opioid withdrawal symptoms between 10 and 12 weeks after treatment initiation. We begin by describing the primary and auxiliary datasets and the causal quantity of interest, followed by estimates obtained under three approaches: (i) using only primary data, (ii) incorporating auxiliary data with known outcome linkage (Assumption A.5(a)), and (iii) incorporating auxiliary data under partial knowledge of the link (Assumption A.5(b)).

## 5.1 Data Description

**Primary Study: XBOT Trial.** The NIDA CTN-0051 (XBOT) trial was a multisite study comparing extended-release naltrexone (XR-NTX) and buprenorphine-naloxone (BUP-NX) for opioid use disorder treatment (Lee et al., 2018). A total of 540 patients were randomized 1:1 to receive either treatments over 24 weeks. We focus on the most severe withdrawal symptoms in the $4^{th}$ week, measured by the Subjective Opiate Withdrawal Scale (SOWS) – a 16-item self-report instrument where patients rate each symptom from 0 to 4, reflecting subjective withdrawal experiences.

**Auxiliary Study: POAT Study.** The NIDA CTN-0030 (POATS) trial enrolled individuals dependent on prescription opioids for outpatient treatment using BUP-NX (Weiss et al., 2011). Withdrawal symptoms were assessed using the Clinical Opiate Withdrawal Scale (COWS), an 11-item clinician-administered tool capturing objective signs of withdrawal. We use POATS as

auxiliary data to improve the estimation of withdrawal severity under BUP-NX in the XBOT trial, leveraging the worst COWS scores in the $4^{th}$ week. Although the XT-NTX arm is absent in the auxiliary dataset, the BUP-NX treatment is shared across both studies. We aim to evaluate if and when auxiliary information on BUP-NX can be used to improve the efficiency of estimating the outcome under BUP-NX in the primary population, while carefully considering the assumptions required for valid data fusion.

## 5.2 Analysis

We evaluate the comparative effectiveness of XR-NTX ($T = 1$) versus BUP-NX ($T = 0$) in reducing withdrawal symptom severity, as measured by the worst SOWS score in the fourth week ($Y$), among participants in the XBOT trial. In the auxiliary POAT study, withdrawal severity is measured on the COWS scale during the same period ($W$). We use a common set of covariates assessed in both trials ($X$). Further, we only considered patients for whom we observed the outcomes – our study excluded individuals for whom treatment was not initiated or who dropped out before our outcome window.

To harmonize the two scales, we derive the transformation coefficient $\alpha$ from published clinical thresholds. According to Wesson and Ling (2003), COWS ranges of 5–12, 13–24, 25–36, and >36 correspond to mild, moderate, moderately severe, and severe withdrawal, respectively. Similarly, Handelsman et al. (1987) defines SOWS ranges of 1–10, 11–15, 16–20, and 21–30 for the same categories. Assuming both scales share a zero point (no withdrawal), we align category midpoints and estimate a linear mapping $Y = \alpha W + \varepsilon$, yielding $\alpha = 0.61$ and intercept $\beta = 0$. Figure 1(a) visualizes this relationship. We assume $\alpha$ is constant across covariate values $X$, and interpret lower values of both $Y$ and $W$ as indicating better outcomes. We then apply the three estimators introduced in Section 4.3 and 4.4: $\hat{\theta}_0$ (primary data only), $\hat{\theta}_b$ (auxiliary data, unknown $\alpha$), and $\hat{\theta}_a$ (auxiliary data, known $\alpha$). As shown in Figure 1(b), $\hat{\theta}_0$ and $\hat{\theta}_b$ suggest that XR-NTX and BUP-NX are almost equally effective. However, $\hat{\theta}_a$ suggests BUP-NX is marginally more effective in lowering withdrawal symptoms compared to XR-NTX. Specifically: (i) $\hat{\theta}_0 = -0.18$ (95% CI width: 1.52), (ii) $\hat{\theta}_b = 0.42$ (95% CI width: 0.65), and (iii) $\hat{\theta}_a = -1.08$ (95% CI width: 1.41). While $\hat{\theta}_a$ achieves a statistically significant result, it relies on the correctness of the assumed $\alpha$. To assess robustness, we conduct a sensitivity analysis by varying $\alpha$ within $\pm 50\%$ of the estimated value, i.e., $\alpha \in [0.31, 0.92]$, assuming the linear form remains valid. Figure 1(c) displays the resulting $\hat{\theta}_a$ estimates across this range. Although the point estimates vary – from $0.50$ to $0.33$ – they consistently favor BUP-NX over XR-NTX. However, for $\alpha > 0.75$, the 95% confidence intervals include zero.

**Takeaways.** In our case study, we assess whether XT-NTX is more effective than BUP-NX in reducing withdrawal symptom severity. The point estimates from the primary study are slightly negative, suggesting a marginal advantage for XT-NTX, but the 95% confidence interval includes zero, indicating no statistically significant difference between the two treatments (see Figure 1(b); estimate $\hat{\theta}_0$). To improve estimation precision, we explore leveraging auxiliary data. Under the strong assumption, our combined analysis yields a statistically significant result favoring BUP-NX over XT-NTX (see Figure 1(b); estimate $\hat{\theta}_a$). While such findings may appear actionable, they hinge critically on an untestable assumption linking outcomes Y and W. If this assumption is violated, the resulting estimates may be misleadingly precise. Our case study thus serves as a *cautionary example*: although integrating auxiliary data can improve precision, it must be done with scrutiny of the underlying assumptions, which—if invalid—can lead to confidently incorrect conclusions.

## 6  Discussion & Conclusion

**Summary.** This paper presents a principled framework for integrating primary and auxiliary datasets with non-overlapping, disparate outcomes to improve efficiency in causal effect estimation. We focus on settings where the primary outcome is never jointly observed with the auxiliary outcome, and we introduce a structural assumption that links the two. Building on this, we define three scenarios reflecting varying levels of prior knowledge about outcome relationship and derive semiparametric efficiency bounds under each. Our findings show that efficiency gains are guaranteed only under the strongest assumptions, when the linking equation is fully known. In contrast, under weaker assumptions, asymptotic efficiency is not ensured. However, finite-sample improvements are still possible, particularly when the auxiliary outcome is highly predictive of the primary outcome. These

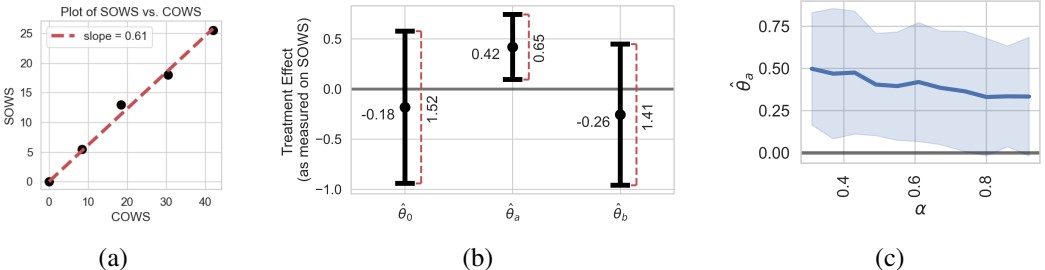

Figure 1: *MOUD Results.* (a) Scatter plot showing the relationship between SOWS and COWS. (b) Treatment effect estimates of MOUD on withdrawal symptoms. Point estimates and corresponding 95% confidence intervals for $\hat{\theta}_0$, $\hat{\theta}_a$ and $\hat{\theta}_b$. (c) Assessing the sensitivity of $\hat{\theta}_a$ to the different values of $\alpha$ in the range 50% above and below the original guess of $\alpha = 0.61$.

benefits taper off as the primary sample size increases, highlighting the limitations of auxiliary data in isolation. We support our theoretical results with both simulations and a case study estimating the effect of medications for opioid use disorder (MOUD) on withdrawal severity. Here, we combine data from the XBOT trial (SOWS scale) and the POAT study (COWS scale), demonstrating the framework's practical utility.

**Limitations & Future works.** Our analysis and results in this paper depend on the structural assumption between $Y$ and $W$. While moving ahead, we will focus on making our results more general by relaxing this assumption; it is important to note that the lack of efficiency gains in a restrictive context would imply a similar conclusion in a more complex context. Further, we will focus on incorporating a third "bridge" dataset, where both outcomes are observed, which could help relax strong assumptions and expand the conditions under which efficiency gains are possible. We will also explore relaxing the assumption of conditional study exchangeability, extending the framework to accommodate discordance in treatments and covariates across studies. Further, our framework requires that at least one treatment arm be shared across the primary and auxiliary studies. When there is no treatment overlap between datasets and the outcomes vary from one study to another, it becomes impossible to link the potential outcome distributions. This makes it difficult to use auxiliary data to enhance efficiency gains. This situation reveals a significant structural limitation in data fusion settings. In the future, we plan to explore data fusion in such scenarios by imposing a distance metric to the treatment space, which will enable us to compare different treatments.

### Acknowledgments

The authors would like to thank the reviewers, the area chair, and the program chair of NeurIPS 2025 for their constructive input to help improve the paper. Further, Harsh Parikh, Kara Rudolph, and Elizabeth Stuart would like to acknowledge that this work was funded by NIH NIDA R01DA056407, and Caleb Miles and Kara Rudolph would like to acknowledge that this work was funded by NIH NIDA R01DA059824. Trang Nguyen and Elizabeth Stuart were funded by NIH NIMH R01MH126856.

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

## Appendix A    Synthetic Data Study and Results

In this section, we are interested in understanding the performance of estimators under various sets of assumptions (A.5(a) to A.5(c)). In particular, we are interested in understanding the potential gains as (i) total number of units, $n$, increases, (ii) the dimensionality of $X$, denoted as $p$, increases and (iii) the log of the ratio of number of units in the auxiliary to primary dataset, $\log\left(\frac{P(S=1)}{P(S=0)}\right)$, increases. First, we discuss our data generative procedures, and then we present and discuss our results.

**Data Generative Procedure.**    The data generation procedure (DGP) in this study is designed to simulate a complex causal structure. We begin by generating covariates $X = (X_1, X_2, \ldots, X_p)$ from a multivariate normal distribution with zero mean and identity covariance matrix where $p$ is the number of covariates. The binary study indicator $S$ is then generated as a Bernoulli random variable, where the probability of assignment to the auxiliary study (i.e., $S = 1$) is $\Pr(S = 1|X) = \text{expit}(a_0 + a_1 X_1 + a_2 X_2)$, where $\text{expit}(x) = \frac{1}{1+e^{-x}}$. The treatment assignment $T$ is also generated as a study and covariate-dependent Bernoulli variable $\Pr(T = 1|X, S) = (1 - S) \times 0.5 + S \times \text{expit}(\zeta_1 X_1)$. The auxiliary outcome $W$, observed only in the auxiliary study ($S = 1$), is defined as follows:

$$W = \mu_W(X, T, S) + \delta = (\gamma_0 + \gamma_1 X_1 + \gamma_2 X_2) \cdot T + \beta_1 X_1 + \beta_2 X_2 + \beta_3 X_3 + \beta_0 + \delta,$$

where vectors $\gamma$ and $\beta$ define the treatment and baseline effects on $W$. This equation includes both linear and interaction terms, capturing treatment-covariate dependencies. In the primary study (where $S = 0$), the primary outcome $Y$ is modeled as: $Y = \alpha(X) \cdot \mu_W(X, T, S) + \gamma$ where $\alpha(X) = \rho_1 X_1 + \rho_0$. This outcome depends on the treatment effect modulated by covariate-driven heterogeneity in $\alpha(X)$, capturing treatment-mediated effects of covariates on $Y$.

Here the true ATE in primary is given as $\theta_0 = \mathbb{E}\left[(\rho_0 + \rho_1 X_1) \cdot (\gamma_0 + \gamma_1 X_1 + \gamma_2 X_2) \mid S = 0\right]$.

**Analysis and Results.**    We use mean-squared error (MSE) to compare the performance of the following three estimators: (i) efficient estimator only using primary data ($\hat{\theta}_0$), (ii) efficient estimator augmented with auxiliary score ($\hat{\theta}_b$) and (iii) efficient estimator with known $\alpha$ integrating auxiliary data ($\hat{\theta}_a$). The simulation results are compiled in Figure 2. As expected, the performance of all three estimators improves as $n$ increases and deteriorates as $p$ increases. Further, $\hat{\theta}_a$ dominates $\hat{\theta}_0$ and $\hat{\theta}_b$ especially for scenarios with large $p$ and/or large $\log\left(\frac{P(S=1)}{P(S=0)}\right)$ – indicating that in scenarios where the primary study is relatively small and the problem is high-dimensional leveraging auxiliary data yields more benefits. This aligns with our theoretical results showing that knowing $\alpha$ can yield efficiency gains. For, $\hat{\theta}_b$ (which uses auxiliary data), we observe that it yields benefits relative to $\hat{\theta}_0$ in small $n$ scenarios especially when $p$ and $\log\left(\frac{P(S=1)}{P(S=0)}\right)$ are large. However, these benefits diminish relative to $\hat{\theta}_0$ as $n$ grows. This is consistent with our theoretical result showing that there are no asymptotic benefits if $\alpha$ is unknown. However, there are some finite sample benefits of using the auxiliary score even when $\alpha$ is unknown.

## Appendix B    Efficiency Score Functions Derivation (Theorems 1–4)

Following the above mentioned procedure we derive the EIFs and corresponding efficiency bounds under the three sets of assumptions. As $\psi^*(O; \theta_0, \eta_0) = \left\{\mathbb{E}[R^*(O; \theta_0, \eta_0) R^*(O; \theta_0, \eta_0)^T]\right\}^{-1} R^*(O; \theta_0, \eta_0)$, and the semiparametrically efficient asymptotic variance (i.e., efficiency bound) is equal to $\left\{\mathbb{E}[R^*(O; \theta_0, \eta_0) R^*(O; \theta_0, \eta_0)^T]\right\}^{-1}$, we only present the efficient score function $R^*$ instead of the EIF $\psi^*$. However, note that deriving the EIF from $R^*$ is straightforward in our context.

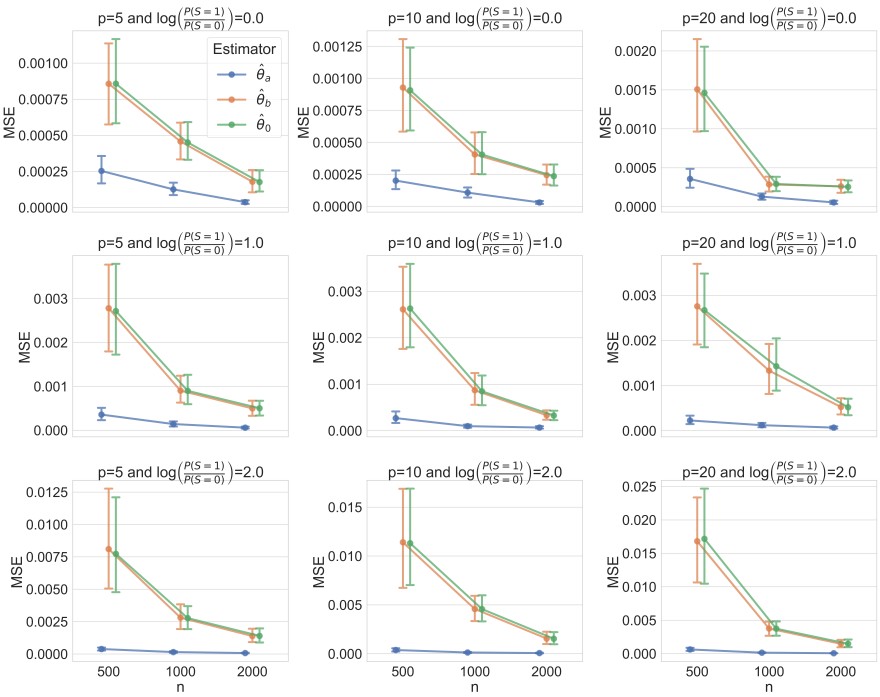

Figure 2: *Simulation Study Results.* Mean squared error rates for three different estimators $\hat{\theta}_0$, $\hat{\theta}_a$ and $\hat{\theta}_b$ based on $R_0^*$, $R_a^*$, and $R_b^*$

In our case, $O = (X, S, T, V)$, $P(O = o; \theta, \eta) = P(X = x)P(S = s \mid X = x)P(T = t \mid X = x, S = s)P(V = v \mid T = t, X = x, S = s)$ and $\mathcal{L}(O; \theta, \eta) = log P(O = o; \theta, \eta)$. Thus,

$$\begin{aligned}
\mathcal{L}(O; \theta, \eta) = & \, log P(X = x) + log P(S = s \mid X = x) + log P(T = t \mid S = s, X = x) \\
& + log P(V = v \mid T = t, S = s, X = x) \\
= & \, log P(X = x) + log(S\mu_S(x) + (1 - S)(1 - \mu_S(x))) \\
& + log(T\mu_T(x, s) + (1 - T)(1 - \mu_T(x, s))) \\
& + log(P(V = v \mid T = t, S = s, X = x)),
\end{aligned}$$

We know that $V = SW + (1 - S)Y$ and $Y = \alpha(X)W + \beta(X) + \varepsilon$.
Thus,

$$P(V = v \mid T = t, S = s, X = x) = P(S(Y - \beta(X) - \varepsilon) + \alpha(X)(1 - S)Y = \alpha(X)v \mid T = t, S = s, X = x).$$

. Simplifying it further,

$$P(V = v \mid T = t, S = s, X = x) = P((S + \alpha(X)(1 - S))Y - S\varepsilon = \alpha(X)v + \beta(X)S \mid T = t, S = s, X = x).$$

Substituting $Y$ with $\theta(X)T + g(X) + \gamma$:

$$\begin{aligned}
& P(V = v \mid T = t, S = s, X = x) \\
& = P((S + \alpha(X)(1 - S))(\theta(X)T + g(X) + \gamma) - S\varepsilon = \alpha(X)v + \beta(X)S \mid T = t, S = s, X = x) \\
& = sP(\gamma - \varepsilon = \alpha(X)(v - \mu_W(X, 1)) - \theta(X)(T - \mu_T(X, 1)) \mid T = t, S = s, X = x) \\
& + (1 - s)P(\gamma = (v - \mu_Y(X, 0)) - \theta(X)(T - \mu_T(X, 0)) \mid T = t, S = s, X = x) \\
& = sP(\alpha(X)\delta = \alpha(X)(v - \mu_W(X, 1)) - \theta(X)(T - \mu_T(X, 1)) \mid T = t, S = s, X = x) \\
& + (1 - s)P(\gamma = (v - \mu_Y(X, 0)) - \theta(X)(T - \mu_T(X, 0)) \mid T = t, S = s, X = x) \\
& = sP(\delta = (v - \mu_W(X, 1)) - \frac{\theta(X)}{\alpha(X)}(T - \mu_T(X, 1)) \mid T = t, S = s, X = x) \\
& + (1 - s)P(\gamma = (v - \mu_Y(X, 0)) - \theta(X)(T - \mu_T(X, 0)) \mid T = t, S = s, X = x).
\end{aligned}$$

Assuming $\gamma$ and $\delta$ are normally distributed with mean 0 and homoskedastic variances $\sigma_\gamma^2$ and $\sigma_\delta^2$ respectively,

$$log P(V = v \mid T = t, S = s, X = x)$$

$$= log \left( \begin{array}{l} s \exp\left( -\frac{((v - \mu_W(x,1)) - \frac{\theta(x)}{\alpha(x)}(t - \mu_T(x,1)))^2}{2\sigma_\delta^2} \right) \\ + (1 - s) \exp\left( -\frac{((v - \mu_Y(x,0)) - \theta(x)(t - \mu_T(X,0)))^2}{2\sigma_\gamma^2} \right) \end{array} \right)$$

**Efficient score function using only primary data (Result of Theorem 1).** Now, we first show the efficient score function for the case that only uses primary data. This works as a baseline case for us and the subsequent efficiency bounds are compared with this case. The efficient score function under assumptions A.2. and A.4. is given as:

$$R_0^*(O; \theta_0, \eta_0) = (1 - S) \cdot \left( ((V - \mu_Y(X,0)) - \theta(X)(T - \mu_T(X,0))) \cdot \frac{(T - \mu_T(X,0))}{\sigma_\gamma^2} \right).$$

Let $\Delta_0 = \left( ((V - \mu_Y(X,0)) - \theta(X)(T - \mu_T(X,0))) \cdot \frac{(T - \mu_T(X,0))}{\sigma_\gamma^2} \right)$ then
$\mathbb{E}\left[ (R_0^*(O; \theta_0, \eta_0))(R_0^*(O; \theta_0, \eta_0))^T \right] = \mathbb{E}\left[ (1 - S)^2 \Delta_0^2 \right]$, and the asymptotic variance

$$\mathbb{V}_0^\theta(X) := \left( \mathbb{E}\left[ (R_0^*(O; \theta_0, \eta_0))(R_0^*(O; \theta_0, \eta_0))^T \right] \right)^{-1} = \frac{1}{\mathbb{E}\left[ \Delta_0^2 \mid S = 0, X \right] p(S = 0 \mid X)}.$$

**Efficient score function under assumptions A.1.–A.4. and A.5(a) (Result of Theorem 2).** Under this assumption, only $\theta$ is unknown and would be estimated using the data while $\alpha$ and $\beta$ are known a priori. Thus, the efficient score function under A.5(a) is:

$$R_a^*(O; \theta_0, \eta_0) = \left( \begin{array}{l} S \cdot \left( (\alpha(X)(V - \mu_W(X,1)) - \theta(X)(T - \mu_T(X,1))) \cdot \frac{(T - \mu_T(X,1))}{\alpha^2(X)\sigma_\delta^2} \right) \\ + (1 - S) \cdot \left( ((V - \mu_Y(X,0)) - \theta(X)(T - \mu_T(X,0))) \cdot \frac{(T - \mu_T(X,0))}{\sigma_\gamma^2} \right) \end{array} \right)$$

Let $\Delta_1 = \left( (\alpha(X)(V - \mu_W(X,1)) - \theta(X)(T - \mu_T(X,1))) \cdot \frac{(T - \mu_T(X,1))}{\alpha^2(X)\sigma_\delta^2} \right)$. Then, the asymptotic variance

$$\mathbb{V}_a^\theta(X) := \left( \mathbb{E}[R_a^*(O; \theta_0, \eta_0)(R_a^*(O; \theta_0, \eta_0))^T \mid X] \right)^{-1} = \left( \begin{array}{l} \mathbb{E}\left[ \Delta_0^2 \mid S = 0, X \right] p(S = 0 \mid X) \\ + \mathbb{E}\left[ \Delta_1^2 \mid S = 1, X \right] p(S = 1 \mid X) \end{array} \right)^{-1}.$$

$\mathbb{V}_a^\theta(X)$ is always smaller than or equal to $\mathbb{V}_0^\theta(X)$ because $\mathbb{E}\left[ \Delta_1^2 \mid S = 1, X \right] p(S = 1 \mid X)$ is non-negative.

**Efficient score function under assumptions A.1.–A.4. and A.5(b) (Result of Theorem 3).** Here, along with $\theta$, $\alpha$ is an unknown parameter. Thus, the efficient score function is given as:

$$R_b^*(O; \{\theta_0, \alpha_0\}, \eta_0) = \left( \begin{array}{l} \left( \begin{array}{l} S \cdot \left( (\alpha(X)(V - \mu_W(X,1)) - \theta(X)(T - \mu_T(X,1))) \cdot \frac{(T - \mu_T(X,1))}{\alpha^2(X)\sigma_\delta^2} \right) \\ + (1 - S) \cdot \left( ((V - \mu_Y(X,0)) - \theta(X)(T - \mu_T(X,0))) \cdot \frac{(T - \mu_T(X,0))}{\sigma_\gamma^2} \right) \end{array} \right) \\ S \cdot \left( \frac{(\theta(X)(T - \mu_T(X,1)) - \alpha(X)(V - \mu_W(X,1)))}{\alpha^2(X)\sigma_\delta^2} \cdot \frac{\theta(X)(t - \mu_T(X,1))}{\alpha(X)} \right) \end{array} \right).$$

$$\mathbb{E}[(R_b^*(R_b^*)^T) \mid X] = \mathbb{E}\left[ \left( \begin{array}{c} S\Delta_1 + (1 - S)\Delta_0 \\ \frac{\theta(X)}{\alpha(X)} S\Delta_1 \end{array} \right) \left( S\Delta_1 + (1 - S)\Delta_0 \quad \frac{\theta(X)}{\alpha(X)} S\Delta_1 \right) \mid X \right]$$

$$= \left( \begin{array}{cc} \mathbb{E}[\Delta_1^2 \mid X, S = 1]P(S = 1 \mid X) + \mathbb{E}[\Delta_0^2 \mid X, S = 0]P(S = 0 \mid X) & \frac{\theta(X)}{\alpha(X)}\mathbb{E}[\Delta_1^2 \mid X, S = 1]P(S = 1 \mid X) \\ \frac{\theta(X)}{\alpha(X)}\mathbb{E}[\Delta_1^2 \mid X, S = 1]P(S = 1 \mid X) & \frac{\theta^2(X)}{\alpha^2(X)}\mathbb{E}[\Delta_1^2 \mid X, S = 1]P(S = 1 \mid X) \end{array} \right)$$

The asymptotic variance-covariance is then

$$\boldsymbol{\Sigma}_b(X) := \left( \begin{array}{cc} \mathbb{V}_b^\theta(X) & Cov_b^{\theta,\alpha}(X) \\ Cov_b^{\theta,\alpha}(X) & \mathbb{V}_b^\alpha(X) \end{array} \right) := \left( \mathbb{E}[(R_b^*(R_b^*)^T) \mid X] \right)^{-1}$$

$$= \left( \begin{array}{cc} \frac{1}{\mathbb{E}[\Delta_0^2 \mid X, S=0]P(S=0 \mid X)} & -\frac{\alpha_0(X)}{\theta_0(X)} \frac{1}{\mathbb{E}[\Delta_0^2 \mid X, S=0]P(S=0 \mid X)} \\ -\frac{\alpha_0(X)}{\theta_0(X)} \frac{1}{\mathbb{E}[\Delta_0^2 \mid X, S=0]P(S=0 \mid X)} & \left( \frac{\alpha_0(X)}{\theta_0(X)} \right)^2 \left( \frac{1}{\mathbb{E}[\Delta_0^2 \mid X, S=0]P(S=0 \mid X)} + \frac{1}{\mathbb{E}[\Delta_1^2 \mid X, S=1]P(S=1 \mid X)} \right) \end{array} \right)$$

From this, we see that the asymptotic variance for the efficient estimator of $\theta$ is $\mathbb{V}_b^\theta(X) = \frac{1}{\mathbb{E}[\Delta_0^2|X,S=0]P(S=0|X)}$. Note, that this asymptotic variance $\mathbb{V}_b^\theta(X) = \mathbb{V}_0^\theta(X)$. This highlights that under assumption $A.5(b)$ there are no efficiency gains from leveraging auxiliary data compared to the baseline which only uses the primary study.

**Efficient score function under assumptions A.1.–A.4. and A.5(c) (Result of Theorem 4).** As the likelihood is agnostic of $\beta$, the efficient score function under A.5(c) is identical to that of A.5(b), i.e.,

$$R_c^*(O; \{\theta_0, \alpha_0\}, \eta_0) = R_b^*(O; \{\theta_0, \alpha_0\}, \eta_0).$$

As the score functions are identical under assumptions A.5(b) and A.5(c), the asymptotic variance is also identical. This indicates that there are no efficiency gains from leveraging auxiliary data compared to the baseline that uses only the primary study.

## Appendix C   Proof of Theorem 5 (Misspecification Bias)

*Proof of Theorem 5.* We begin by defining $\hat{\theta}_a$ as the estimator solving the empirical moment condition $\mathcal{P}_n R_a^*(O; \hat{\theta}_a, \hat{\eta}) = 0$. In the population, $\theta_0$ solves $\mathbb{E}[R_a^*(O; \theta_0, \eta_0)] = 0$ only under the correct specification of $\alpha = \alpha^\star$. We now investigate what happens when the analyst assumes $\alpha = \alpha_{\text{mis}}$, where $\alpha_{\text{mis}} \neq \alpha^\star$. Recall, $\hat{\theta}_a$ is

$$\frac{\sum_i \left( (1 - S_i)\frac{\hat{r}_Y(X_i,0)\hat{r}_T(X_i,0)}{\hat{\sigma}_Y^2} + S_i \frac{\hat{r}_W(X_i,1)\hat{r}_T(X_i,1)}{\alpha(X_i)\hat{\sigma}_W^2} \right)}{\sum_i \left( (1 - S_i)\frac{\hat{r}_T^2(X_i,0)}{\hat{\sigma}_Y^2} + S_i \frac{\hat{r}_T^2(X_i,1)}{\alpha^2(X_i)\hat{\sigma}_W^2} \right)}$$

Thus, $\mathbb{E}[\hat{\theta}_a(\alpha_{mis}) - \hat{\theta}_a(\alpha^\star)] = \mathbb{E}[\hat{\theta}_a(\alpha_{mis}) - \hat{\theta}_a(\alpha^\star) \mid S = 0]P(S = 0) + \mathbb{E}[\hat{\theta}_a(\alpha_{mis}) - \hat{\theta}_a(\alpha^\star) \mid S = 1]P(S = 1)$. In the estimator, terms with $(1 - S)$ do not interact with $\alpha$. Thus, $\mathbb{E}[\hat{\theta}_a(\alpha_{mis}) - \hat{\theta}_a(\alpha^\star) \mid S = 0]P(S = 0) = 0$. Now, consider $\mathbb{E}[\hat{\theta}_a(\alpha_{mis}) - \hat{\theta}_a(\alpha^\star) \mid S = 1]P(S = 1)$.

$$
\begin{aligned}
\mathbb{E}[\hat{\theta}_a(\alpha_{mis}) - \hat{\theta}_a(\alpha^\star) \mid S = 1] &= \mathbb{E}[\mathbb{E}[\hat{\theta}_a(\alpha_{mis}) - \hat{\theta}_a(\alpha^\star) \mid X, S = 1] \mid S = 1] \\
\mathbb{E}[\hat{\theta}_a(\alpha_{mis}) - \hat{\theta}_a(\alpha^\star) \mid X, S = 1] &= \mathbb{E}\left[ \frac{(\alpha_{mis}(X) - \alpha^\star(X))\mathbb{E}[\hat{r}_W(X,1)\hat{r}_T(X,1)]}{(\mathbb{E}[\hat{r}_T^2(X,1)])} \mid X, S = 1 \right] \\
&= \mathbb{E}\left[ \frac{(\alpha_{mis}(X) - \alpha^\star(X))}{\alpha^\star(X)} \frac{\mathbb{E}[\hat{r}_Y(X,1)\hat{r}_T(X,1)]}{(\mathbb{E}[\hat{r}_T^2(X,1)])} \mid X, S = 1 \right] \\
&= \mathbb{E}\left[ \frac{(\alpha_{mis}(X) - \alpha^\star(X))}{\alpha^\star(X)}\theta(X) \mid X, S = 1 \right]
\end{aligned}
$$

$\square$

## Appendix D   Proof of Theorem 6 (Error bound for $\hat{\mu}_Y$)

*Proof.* We analyze the estimation error for both $\hat{\mu}_{Y,0}$ and $\hat{\mu}_{Y,b}$ under the given metric entropy assumptions.

**(i) One-stage estimator $\hat{\mu}_{Y,0}$.** By assumption A.6., $\mu_Y \in \mathcal{M}$, and the metric entropy of $\mathcal{M}$ satisfies

$$\log N(\varepsilon, \mathcal{M}, \|\cdot\|) \leq C\varepsilon^{-\omega}.$$

From standard results in empirical process theory and nonparametric regression (e.g., Györfi et al. (2006) and Tsybakov and Tsybakov (2009)), it follows that the least-squares estimator $\hat{\mu}_{Y,0}$ satisfies

$$\|\hat{\mu}_{Y,0} - \mu_Y\| = o_p\left( n_0^{-1/(2+\omega)} \right).$$

**(ii) Two-stage estimator $\hat{\mu}_{Y,b}$.** By assumption A.5., $\mu_Y(X) = \alpha(X)\mu_W(X) + \beta(X)$. We estimate $\hat{\mu}_W(X)$ from $n_1$ auxiliary samples. Let $\hat{\mu}_W$ be an estimator satisfying

$$\|\hat{\mu}_W - \mu_W\| = o_p\left(n_1^{-1/(2+\omega)}\right),$$

under the assumption that $\mu_W \in \mathcal{M}$ and satisfies the same entropy bound as $\mu_Y$.

The two-stage estimator is defined as:

$$\hat{\mu}_{Y,b}(X) = \hat{\alpha}(X) \cdot \hat{\mu}_W(X, 0) + \hat{\beta}(X),$$

where $(\hat{\alpha}, \hat{\beta})$ minimize the squared error loss over the primary sample:

$$(\hat{\alpha}, \hat{\beta}) = \arg\min_{\alpha \in \mathcal{A},\ \beta \in \mathcal{B}} \frac{1}{n_0} \sum_{i=1}^{n_0} (Y_i - \alpha(X_i)\hat{\mu}_W(X_i, 0) - \beta(X_i))^2.$$

We now decompose the error:

$$\|\hat{\mu}_{Y,b} - \mu_Y\| = \|\hat{\alpha}\hat{\mu}_W + \hat{\beta} - \alpha\mu_W - \beta\|.$$

Adding and subtracting intermediate terms:

$$= \|(\hat{\alpha} - \alpha)\hat{\mu}_W + \alpha(\hat{\mu}_W - \mu_W) + (\hat{\beta} - \beta)\|.$$

Applying triangle inequality:

$$\|\hat{\mu}_{Y,b} - \mu_Y\| \leq \|(\hat{\alpha} - \alpha)\|\|\hat{\mu}_W\|_\infty + \|\alpha\|_\infty\|(\hat{\mu}_W - \mu_W)\| + \|\hat{\beta} - \beta\|.$$

Under the assumption that $\hat{\mu}_W$ is uniformly bounded (which holds if $\mu_W$ and $\hat{\mu}_W$ are bounded and consistent), and using the entropy conditions on $\mathcal{A}$ and $\mathcal{B}$:

$$\|\hat{\alpha} - \alpha\| = o_p(n_0^{-1/(2+\omega_\alpha)}) \quad \text{(given A.6.), and,} \quad \|\hat{\beta} - \beta\| = 0 \quad \text{(given A.5(b)).}$$

Combining all the pieces, we obtain:

$$\|\hat{\mu}_{Y,b} - \mu_Y\| = o_p\left(n_0^{-1/(2+\omega)}\left(n_0^{\frac{1}{2+\omega} - \frac{1}{2+\omega_\alpha}} + \left(\frac{n_1}{n_0}\right)^{-1/(2+\omega)}\right)\right),$$

where the first term reflects the complexity reduction from modeling $\mu_Y$ via $\alpha(X)$, and the second term reflects the error propagated from estimating $\mu_W$ using auxiliary data. $\qquad\square$

