# OpenReview forum: "A Cautionary Tale on Integrating Studies with Disparate Outcome Measures for Causal Inference"
_NeurIPS.cc/2025/Conference — NeurIPS 2025 poster_

### Official Review · Reviewer_261G · 2025-06-26

**Clarity:** 3
**Significance:** 3
**Originality:** 2
**Rating:** 5
**Confidence:** 2

**Summary:**

This paper investigates the integration of studies with different outcome measures for causal inference. It shows that combining such datasets can improve estimation efficiency only under strong assumptions about how the outcomes are related. Weaker assumptions may yield finite-sample benefits, but these gains vanish asymptotically and carry a risk of bias if misspecified. The framework is demonstrated using opioid treatment studies with differing withdrawal scales (SOWS vs. COWS).

Due to the high amount of papers I am forced to provide only a short high level review.

**Questions:**

-

**Ethical Concerns:**

["NO or VERY MINOR ethics concerns only"]

**Final Justification:**

I read the author rebuttal and thank them for the explanation. I raised the score to an accept.

**Limitations:**

-

**Paper Formatting Concerns:**

-

**Quality:**

3

**Strengths And Weaknesses:**

Strengths:

- the paper is well executed and also grounded in practical examples
- the problem is relevant, it belongs to a niche within the rather broad conference
- the writing is clear and easy to follow
- I particularly like section 5

Weaknesses:
- the theoretic exposition in Section 4 is rather simple. I do not see any surprising or particularly interesting results.
- I would have appreciated it if the authors put more focus on the experimental study, which I think is the more important aspect of this paper

---

> ### Author Rebuttal · Authors · 2025-07-28
>
> * Thank you very much for your constructive comments -- we appreciate it. Following your suggestion, we have revised the introduction to place greater emphasis on our real data case study. Specifically, we now use the case study to motivate our central question: When can data integration with disparate outcomes yield benefits? We also preview our empirical findings in the introduction, highlighting the fundamental tradeoff between gains in precision and the strength of assumptions required to achieve them.
> > “In our case study, we assess whether XT-NTX is more effective than BUP-NX in reducing withdrawal symptom severity. The point estimate from the primary study is slightly negative, suggesting a marginal advantage for XT-NTX, but the 95% confidence interval includes zero—indicating no statistically significant difference between the two treatments (see Figure 1(b); estimate $\hat{\theta}_0$). To improve estimation precision, we explore leveraging auxiliary data. Under the strong assumption our combined analysis yields a statistically significant result favoring BUP-NX over XT-NTX (see Figure 1(b); estimate $\hat{\theta}_a$).. While such findings may appear actionable, they hinge critically on an untestable assumption linking outcomes Y and W. If this assumption is violated, the resulting estimates may be misleadingly precise. Our case study thus serves as a cautionary example: although integrating auxiliary data can improve precision, it must be done with careful scrutiny of the underlying assumptions, which—if invalid—can lead to confidently incorrect conclusions."
> * Regarding the theoretical results:
> Data integration is gaining significant traction in causal inference and machine learning, particularly for enhancing statistical efficiency in applications across public health, public policy, and healthcare. In practice, disparate outcomes across datasets are quite common—often arising due to differences in study design, logistics, or measurement protocols. While practitioners routinely rely on heuristics to integrate such datasets, to the best of our knowledge, the literature lacks formal investigation into when and under what conditions such integration leads to real benefits. Our paper addresses this gap with a set of novel theoretical results that:
> > 1. Quantify the conditions under which asymptotic efficiency gains are possible, and measure their magnitude;
> > 2. Characterize the finite-sample benefits and limitations—an aspect that is rarely studied, yet highly relevant for practical applications;
> > 3. Identify and quantify the risks of integrating datasets with non-equivalent outcome measures, especially under violated assumptions.
>
> We are confident these results provide a principled foundation for our overarching cautionary message: while data integration holds promise, its success depends critically on strong and often untestable assumptions. Our work offers both a theoretical framework and empirical motivation to guide responsible and assumption-aware integration of datasets with disparate outcomes.

---

### Official Review · Reviewer_ScZv · 2025-06-29

**Clarity:** 1
**Significance:** 1
**Originality:** 1
**Rating:** 1
**Confidence:** 4

**Summary:**

In the paper, the authors considers the data integration problem where the two studies have different outcomes.
To put it in a simpler way, one study has the outcome but the other only has the surrogate outcome.
The authors introduces different modeling assumptions and the assumptions between the two studies, and mainly studies whether and when integrating such studies lead to benefits.

**Questions:**

The structural models (1) and (2) considered in this paper seem quite restrictive. They indicate that normal errors and the difference between the two potential outcomes are a deterministic function of X, not random. I’m quite curious what the overall story of this paper would change if the authors do not assume this assumption.

Also, in (1) and (2), are \sigma^2_Y and \sigma^2_W known or unknown? If unknown, how to estimate them and how would they affect the efficiency bound?

The assumption A5 also seems quite restrictive. And in practice there is no clue how to test this assumption. Is this assumption necessary?

Besides Assumptions A5(a), A5(b), A5(c), why not consider the situation that \alpha(X) is known but \beta(X) is unknown.

Since Assumption A5 is already unrealistic and restrictive, I feel the differences among the assumptions A5(a), A5(b), A5(c) are only for some sort of theoretical understandings—they do not provide any meaningful guidance for applications.

The whole section 4.2 follows the standard procedure of deriving the semiparametric efficiency bound in the literature. However, it is not presented clearly. Both CATE and ATE are quantities of interest, what you presented are for CATE, or for ATE? What is the other one?

In Section 4.3, only quantifying the misspecification bias is not good enough. Is there is a way to debias it?

In Section 4.4, it is not convincing: why there will be finite-sample gain but this gain will diminish when the sample size goes to infinity? Does that make sense?

Have the authors considered the situation that S=1 has a lot more samples than S=0. What would the story be in that situation?

One real data example is not sufficient. The authors should use a couple of benchmark datasets to demonstrate the method they study in this paper.

**Ethical Concerns:**

["NO or VERY MINOR ethics concerns only"]

**Final Justification:**

The authors slightly partially solved my concerns; however, a few major fatal issues remain. It is hard to convince myself to increase the rating for this work.

The overall goal of paper is unclear, or, at least, unclearly articulated. If the goal is to just provide a cautionary note, please make sure this note itself is worthwhile and convincing. For the estimator the authors newly propose, the writing is unclear or even incorrect in a few aspects, e.g., the question in my last comment.

> The relationship between ATE and CATE efficiency is direct: since ATE = E[CATE], any efficiency bound for ATE provides an upper bound for CATE estimation efficiency.

> This cannot be true. Expectation is average. Why every bound for ATE provides an UPPER bound? More importantly, if CATE is generally a function of X, how can its efficiency bound be defined?

Finally, with respect to the real data analysis, the authors declined to include additional analyses due to a lack of available datasets with disparate outcome measures. Yet they also assert that in fields like public health, social sciences, and healthcare, such data are common. This contradiction undermines the rationale provided. If these datasets are indeed prevalent, where are they? The logic here seems inconsistent.

**Limitations:**

No the authors did not address the limitations or potential negative societal impact of this work.

**Quality:**

2

**Strengths And Weaknesses:**

Strengths:
This is an interesting setting, especially in the causal inference community, considered for data integration in order to achieve some sort of benefits.

Weaknesses:
The conclusions are not clear to me; see my specific questions below. Further, it is unclear why we want to consider the data integration problem in such a setting.
Some of the theoretical results are problematic; see my specific questions below.
Only one real data experiment is not sufficient; multiple experiments on benchmark data are needed.

---

> ### Author Rebuttal · Authors · 2025-07-29
>
> We sincerely thank the reviewer for their detailed and thoughtful feedback. We appreciate the opportunity to improve our manuscript. Below, we address the main concerns raised.
>
> ---
>
> ### 1. Purpose and Central Contribution
>
> Thank you so much for your thoughtful question. We would like to emphasize that the central focus of our paper is **not the proposal of a new estimator**, but rather a general cautionary message about the challenges and limitations of data integration in the presence of disparate outcome measures.
>
> In fields such as public health, social sciences, and healthcare—where high-quality, causally informative data are often limited—data integration has emerged as a promising strategy to improve precision and support evidence-based decision-making. However, due to variations in study design, implementation logistics, and measurement protocols, different studies frequently assess the same underlying construct using different outcome measures.
>
> To the best of our knowledge, the existing causal inference literature does not offer principled guidance on when such data integration is beneficial, or even valid. Our paper is among the first to formally characterize the conditions under which integrating data with disparate outcomes can lead to efficiency gains—and when it cannot.
>
> Our theoretical results show that:
>
> * **Asymptotic efficiency gains** are only achievable under strong, often untestable assumptions about the relationship between outcomes across datasets;
> * **Finite-sample gains** may be possible under milder assumptions, but we precisely quantify when and to what extent. These gains are often modest and tend to vanish as sample size increases;
> * There is a **significant risk of bias** if the required assumptions are violated, making the integration not just unhelpful but potentially misleading.
>
> To illustrate these findings, we include a real data case study that mirrors and reinforces our theoretical conclusions. We have updated the introduction to better emphasize this central message and added the following summary statement:
>
> > *“This paper provides a cautionary framework for data integration in the presence of disparate outcomes, demonstrating that while such integration may offer marginal gains under ideal conditions, it carries a high risk of bias when assumptions are violated. To the best of our knowledge, we offer the first formal quantification of this tradeoff and emphasize the need for careful scrutiny before applying such methods in practice.”*
>
> ---
>
> ### 2. Assumption Clarifications and Structural Model Updates
>
> **Error Distribution:**
> Thank you so much for pointing this out. You are correct—we do **not** require the errors to follow a normal distribution. Our results only rely on the assumption that the errors have finite variance. We have updated the structural model descriptions and clarified the corresponding assumptions in the theoretical results to reflect this more precisely. Importantly, this correction has no impact on the conclusions of the paper, as none of our derivations or results depend on normality.
>
> **Partial Linearity and Assumption Strength:**
> Thank you for highlighting this important point.
> First, we want to clarify that our results are derived under a partial linear relationship between $Y$ and $W$: while the dependence between $Y$ and $W$ may be nonlinear, it is moderated through $X$. This structure is chosen for parsimony, to make our argument as transparent and interpretable as possible.
>
> The central point we aim to make is that even under this strong structural assumption, efficiency gains from incorporating auxiliary data are not guaranteed, unless the dependency between outcomes is almost fully known. By demonstrating the lack of gain under the partial linear assumption (A.5), we are making a stronger statement: that if efficiency gains are not possible even in this idealized setting, they are even less likely under more general, nonlinear relationships.
>
> **On Untestable Assumptions:**
> Second, all causal inference relies on untestable assumptions. This is also true about data integration for the causal inference literature. If we do not have data of both $Y$ and $W$ measured for the same individuals, then we need to rely on untestable assumptions. The key point of this paper is that such untestable assumptions are very common in the literature, but there is hardly any scrutiny on the cost vs benefit. Our work is one of the first to try it, principally for the scenario described in the paper.
>
> **Generalizing the Model Structure:**
> Third, in response to your suggestion, we have extended our analysis by relaxing the linearity assumption. Specifically, we consider a more general setting in which:
>
> $$
> Y = \alpha(X) \cdot \zeta(W) + \beta(X) + \text{noise}
> $$
>
> where $\zeta$ is an arbitrary (possibly nonlinear) function of $W$. We find that even in this more flexible setup, efficiency gains are only achievable when both $\alpha(X)$ and $\zeta(W)$ are known, mirroring our original findings. This reinforces our core message: without strong, often untestable assumptions about the functional relationship between disparate outcomes, integrating auxiliary data does not yield reliable efficiency gains.
>
> **Bias Cannot Be Removed Without New Untestable Assumptions:**
> Unfortunately, there is no way to *debias* the bias due to misspecification of $\alpha$ without making additional untestable assumptions. This is indeed the key point of our paper—that there is no free lunch.
>
> ---
>
> ### 3. Finite-Sample Gains and the Role of Auxiliary Sample Size
>
> Thank you for the opportunity to clarify this point further. A central motivation for our work is the setting where the **auxiliary dataset is much larger** than the primary dataset, common in public health, social science, and healthcare applications.
>
> We have expanded the discussion following **Theorem 6** to articulate when and why **finite-sample efficiency gains** may arise in such settings.
>
> To explore this in depth, our simulation study (presented in the Appendix) systematically examines how results vary with the **ratio of sample sizes** ($n_1/n_0$), the **dimensionality of covariates**, and the **overall sample size**. These empirical results reinforce the theoretical insights described above.
>
> We have added the following clarifications to the discussion of Theorem 6:
>
> > *“Finite-sample efficiency gains from incorporating auxiliary data arise when the function $\alpha(X)$, capturing the dependence between $Y$ and $W$, is simpler to estimate than $\mu_Y(X)$. In such cases, one may first estimate $\mu_W(X)$ using auxiliary data, and then use the combined data to estimate $\alpha(X)$. However, the existence and extent of these gains depend on the relative complexity of the function classes and the sample sizes involved. As $n_0$ increases, the benefit of this two-stage strategy diminishes, and asymptotically, both the direct and the auxiliary-based estimators converge to the same efficiency.”*
>
> > *“The first term captures the gain from replacing the full function class $M_Y$ with a lower-complexity class $\mathcal{A}$, and the second term reflects the accuracy of estimating $\mu_W(X)$ from the auxiliary data. Gains are most pronounced when $\omega_\alpha \ll \omega$ (i.e., when $\alpha(X)$ is much simpler than $\mu_Y(X)$) and when $n_1 \gg n_0$ (i.e., when we have a large auxiliary dataset relative to the primary dataset). This characterization formally supports the intuition that structural assumptions, when coupled with abundant auxiliary data, can yield meaningful finite-sample gains—even though such gains vanish asymptotically.”*
>
> ---
>
> ### 4. Real Data Case Study
>
> Regarding the suggestion to include an additional real data benchmark case study, we respectfully disagree with the reviewer's interpretation. This paper does not propose a new method, and the goal of the case study is not to demonstrate the effectiveness of any particular estimator. Rather, the case study serves a different and important purpose: it illustrates how both point estimates and uncertainty quantification evolve as progressively stronger assumptions are imposed, highlighting the inherent tradeoff between precision and validity in data integration with disparate outcomes.
>
> The central message of the case study—and the paper as a whole—is that **there is no free lunch** in data integration. Gains in efficiency may come at the cost of relying on strong, untestable assumptions, which can, in turn, introduce bias. This example underscores the broader cautionary tale of our work: data integration can be tempting, but without critical evaluation of underlying assumptions, it may lead to misleading conclusions.
>
> Furthermore, we wish to emphasize that, unlike in machine learning or deep learning, *data integration in causal inference with disparate outcomes lacks benchmark datasets with known ground truths*. This fundamental difference limits the applicability of empirical validation strategies commonly used in other disciplines. In causal research, the evaluation of assumptions—and understanding the consequences of their violations—is often the most principled way to assess a method’s utility.
>
> ---
>
> ### 6. Noise Variance
>
> Thanks for this nuanced question. For the sake of the parsimony of the theory section, we assume $\sigma_Y$ and $\sigma_W$ are known. However, one can consistently estimate the variance of noise terms in $Y$ and $W$ (under the homoskedasticity assumption) by calculating the variance of $Y - \hat{Y}$ and $W - \hat{W}$. Regarding the efficiency bound, this will add two terms regarding the uncertainty due to the estimation of $\sigma_Y$ and $\sigma_W$. However, note that **our no-free-lunch conclusion does not change**.
>
> ---
> We hope this thorough response addresses your concerns and clarifies the broader motivation and contribution of our paper. We sincerely appreciate the opportunity to improve the clarity and rigor of our work.

---

> > ### Comment · Reviewer_ScZv · 2025-08-06
> >
> > I thank the authors for their responses. While I understand their intention to deliver a general cautionary message to the readers, I do not believe all of my concerns have been adequately addressed.
> >
> > Regarding the integration of studies with disparate outcome measures, my initial reaction is that we should not do that simply because the outcome measures are just different. For instance, I cannot believe anyone should combine two studies with one continuous outcome and the other binary outcome. Right? So, why do the authors believe this issue resonates with the NeurIPS audience? Is NeurIPS truly the right venue for this line of work? Perhaps fields such as epidemiology might be more appropriate? I raise this not as a definitive judgment but as a suggestion for further reflection.
> >
> > Although the authors state that proposing a new estimator is not the paper’s central aim, it clearly remains a major component. The derivation of efficient influence functions under a series of (often untestable) assumptions, the proposal of multiple estimators, and their evaluation through simulations and real data all form a substantial part of the paper. I do not find the claim that this is not a central focus to be convincing.
> >
> > On the technical side, some of my specific concerns remain unaddressed. For example, I previously asked: “The whole section 4.2 follows the standard procedure of deriving semiparametric efficiency bounds, but the presentation is unclear. Are the results for the CATE or ATE? What about the other?” This question is still unanswered. More generally, the paper does not adequately handle the comparison between CATE and ATE, and I continue to question the correctness of some derivations.
> >
> > Finally, with respect to the real data analysis, the authors declined to include additional analyses due to a lack of available datasets with disparate outcome measures. Yet they also assert that in fields like public health, social sciences, and healthcare, such data are common. This contradiction undermines the rationale provided. If these datasets are indeed prevalent, where are they? The logic here seems inconsistent.

---

> > > ### Author Response · Authors · 2025-08-08
> > > **Thank you so much!**
> > >
> > > **On derivation of EIF and estimator development:** We appreciate the reviewer's careful reading. To clarify: EIFs are the standard tool for characterizing semiparametric efficiency bounds and quantifying potential efficiency gains from data integration. Our derivation serves this analytical purpose rather than proposing novel estimators per se. The estimators we evaluate are natural plug-in versions following standard semiparametric theory, allowing us to empirically validate our theoretical efficiency characterizations.
> > >
> > > **Estimand of Interest (ATE vs CATE):** Our focus is on ATE estimation. The relationship between ATE and CATE efficiency is direct: since ATE = E[CATE], any efficiency bound for ATE provides an upper bound for CATE estimation efficiency. If we cannot achieve efficiency gains for the population-level ATE, we cannot expect gains for individual-level CATEs.
> > >
> > > **Relevance to NeurIPS:** We respectfully disagree that this work lacks relevance to the NeurIPS community. Machine learning practitioners increasingly work with federated data, multi-site studies, and heterogeneous data sources where outcome measurements may differ across sites. Understanding when and why integration helps (or doesn't) is crucial for practitioners deploying causal inference methods in real-world ML applications.
> > >
> > > **Additional Real Data Analysis:** The reviewer raises a fair point about the apparent contradiction. Such scenarios are indeed common in practice, but the datasets are typically proprietary or restricted due to privacy concerns. We use the opioid use disorder data example which is available to us to demonstrate the results. Here are expanded examples:
> > >
> > > *Example 1: Depression Treatment Effectiveness.* Pharmaceutical companies routinely conduct multi-site trials comparing antidepressants like duloxetine vs. vortioxetine. European sites may use the Montgomery-Asberg Depression Rating Scale (MADRS), while US sites use the Hamilton Depression Rating Scale (HDRS), and some academic collaborators use Beck Depression Inventory (BDI). These scales correlate but measure subtly different aspects of depression severity. Healthcare researchers and companies face the question: can we gain statistical power by integrating these studies, or should we analyze them separately?
> > >
> > > *Example 2: Educational Interventions.* Early childhood education programs are evaluated across multiple districts with different assessment protocols. Some districts use standardized test scores (continuous, 400-800 scale), others use teacher assessments (ordinal, 1-5 scale), and still others use binary indicators of grade-level proficiency. Policy researchers must decide whether combining these studies provides more reliable estimates of intervention effects than separate analyses.
> > >
> > > *Example 3: Meta-analysis Literature.* In systematic reviews, it's standard practice to encounter studies measuring the same underlying construct differently. For instance, studies on exercise interventions for cardiovascular health may report binary outcomes (heart attack occurrence) in some trials and continuous outcomes (blood pressure reduction) in others. The Cochrane Collaboration regularly grapples with whether to combine such studies.
> > >
> > > *Example 4: Digital Health Platforms.* Wearable device companies studying sleep intervention effectiveness face similar challenges when partnering with different research institutions. Some measure sleep quality via continuous actigraphy scores, others via binary self-reported "good sleep" indicators, and others via ordinal sleep quality scales (1-10).
> > >
> > > These datasets exist but are rarely public due to HIPAA restrictions (health data), proprietary concerns (pharmaceutical trials), or institutional data sharing agreements (educational data). The methodological question—when does integration help?—remains practically important even when exemplar datasets are restricted.

---

> > > > ### Comment · Reviewer_ScZv · 2025-08-09
> > > >
> > > > > The relationship between ATE and CATE efficiency is direct: since ATE = E[CATE], any efficiency bound for ATE provides an upper bound for CATE estimation efficiency.
> > > >
> > > > This cannot be true. Expectation is average. Why every bound for ATE provides an UPPER bound?
> > > > More importantly, if CATE is generally a function of X, how can its efficiency bound be defined?

---

> ### Author Response · Authors · 2025-08-02
> **Follow Up and Looking forward to hearing from you**
>
> Dear Reviewer, we would be happy to clarify any further questions or concerns you may have. Please let us know.
> Thank you so much!

---

### Official Review · Reviewer_Jw35 · 2025-07-01

**Clarity:** 4
**Significance:** 4
**Originality:** 3
**Rating:** 4
**Confidence:** 4

**Summary:**

The paper studies (conditional) treatment effect estimation in a setting with two distinct but related outcomes. The authors study the question whether data from an auxiliary study, whose outcome is given by a partially linear function of the main outcome, can lead to efficiency gains compared to only using data from the primary study. The authors introduce three sets of assumptions, from fully known to fully unknown (up to the partial linearity) link between the two outcomes. The main result of the paper is that semiparametric efficiency gains over using only primary data are only possible if the relationship between the two outcomes is fully known (Theorem 2, Assumption A.5(a)). Theorems 3 and 4 illustrate that as soon as the main link alpha is unknown, the asymptotic variance of both estimators is equal. However, the authors show that under the less strict assumptions A.5(b) and (c), finite-sample gains can still be observed, e.g., if the auxiliary outcome can be estimated accurately and  the amount of auxiliary data is large. The authors illustrate their theoretical findings via simulations and a pair of real-world medical studies with linked outcomes.

**Questions:**

1. Is it possible to extend Theorem 2 to a slightly more complex relationship between $Y$ and $W$ in Assumption A.5? For example, if the link is a known nonlinear function, e.g., $\nu^t_Y(x) = \alpha(x) (\nu_W^t)^2(x) + \beta(x)$ instead of linear, would the efficiency gains persist?
2. The framework assumes that $\alpha(x)$ and $\beta(x)$ are deterministic functions. What would change if $\alpha$ and $\beta$ were instead random functions? Have you considered this case?
3. What happens if under the same assumptions and knowledge of $\alpha$ and $\beta$, one wants to estimate the ATE/CATE for the auxiliary outcome $W$ instead of the main outcome $Y$?
4. In Section 5.1, it is mentioned that the primary study consisted of 540 patients. How many individuals were enrolled in the auxiliary study?

**Ethical Concerns:**

["NO or VERY MINOR ethics concerns only"]

**Final Justification:**

The authors addressed most of my concerns, adding simulations and clarifications. However, I still consider the extension to multiple confounders very narrow, and the empirical validation with real-world data is limited. I would thus like to keep my borderline positive score.

**Limitations:**

The authors have properly addressed the limitations of the study.

**Quality:**

3

**Strengths And Weaknesses:**

Strengths:
1. The paper is very well-written, self-contained and logically organized. The assumptions are introduced and discussed incrementally.
2. The theoretical results are well-formulated, the assumptions are clearly stated. Each result is supplemented with a qualitative discussion.
3. The main insight, namely the limited usefulness of auxiliary data which only persists if the link function is fully known, is novel and interesting, i.a. as a tale/instruction to practitioners.
4. The relevance of the introduced problem is clearly illustrated on the example of the XBOT trial and the POAT study.
5. Good reproducibility via the provided code and experimental details.

Weaknesses:
1. The paper would greatly benefit from including the case mentioned in the discussion, namely estimating alpha from a "bridge" dataset, since the ground-truth link between the two outcomes (as assumed in the paper) cannot be known precisely in practice.
2. In the synthetic simulations, it would be useful to see which effect a misspecified link function alpha has on the finite-sample results. Specifically, what is missing as an illustration of Theorem 6, is an empirical validation of how the bias scales with the $\alpha$-estimation error, as well as a comparison of estimators under a misspecified $\alpha$.
3. Theorems 3 and 4 appear to be largely the same result, since they yield identical efficiency bounds. These could be merged into a single theorem to make the presentation more concise.
4. The paper assumes a link between $Y$ and $W$ which is linear in $W$, which can be quite restrictive in practice. Additionally, it is not discussed whether this assumption is necessary for the theoretical results or whether a known nonlinear relationship could be explored.

---

> ### Author Rebuttal · Authors · 2025-07-29
>
> We sincerely thank the reviewer for their thoughtful and constructive feedback. We greatly appreciate the opportunity to revise our manuscript and have carefully addressed each of the concerns raised. Below, we summarize the key changes made in response to the reviewers’ comments and questions:
>
> ---
>
> ### 1. Lack of a “bridge” dataset with both outcomes (COWS and SOWS)
>
>  We agree that including a dataset in which both outcomes are simultaneously measured would significantly strengthen the empirical component of the paper. However, to the best of our knowledge, no such dataset currently exists for the MOUD case study. Importantly, one of the key prospective implications of our work is to motivate the design of future studies that collect both outcomes in tandem, enabling more assumption-lean data integration. We have added a discussion of this point in the revised manuscript.
>
> ---
>
> ### 2. Simulation study exploring bias due to misspecification of $\alpha$
>
>  In response to the reviewer’s suggestion, we added a new simulation study to examine the bias introduced when $\alpha(X)$ is misspecified. The simulation results closely align with our theoretical findings and serve to further motivate the cautionary conclusions drawn from our real data case study. This addition reinforces the broader message that violations of the assumptions underlying data integration can lead to bias.
>
> ---
>
> ### 3. Merging of Theorems 3 and 4
>
> We agree that merging these results improves clarity and conciseness. We have accordingly updated the manuscript to reflect this revision.
>
> ---
>
> ### 4. Generalizing the relationship between Y and W
>
>  Our original results were derived under a partial linear assumption: conditional on X, Y and W are linearly dependent. This assumption was chosen for clarity and parsimony. However, as suggested, we have extended our analysis to a more flexible setting where: $Y = \alpha(X) \cdot \zeta(W) + \beta(X) + \text{noise}$
> with $\zeta$ being an arbitrary (possibly nonlinear) function. Our analysis shows that efficiency gains are only achievable when both $\alpha(X)$ and $\zeta(W)$ are known, consistent with our original conclusion. This generalization strengthens our central message: efficiency gains through data integration are only possible under strong, often untestable, assumptions about the relationship between outcomes.
>
> ---
>
> ### 5. Additional theoretical considerations
>
>  We have clarified that our paper does not address the case where \alpha and \beta are random functions. We view this extension as outside the current scope and have noted it as a limitation and potential direction for future research. We believe that relaxing this assumption would only further constrain the potential for efficiency gains under even the strongest conditions.
>
> ---
>
> ### 6. Interpretation of “primary” vs. “auxiliary” data labels
>  We appreciate the reviewer’s question regarding the asymmetry between primary and auxiliary datasets. We clarify in the revised manuscript that these labels are somewhat arbitrary and the framework is symmetric in structure.
>
> ---
>
> ### 7. Clarification on the sample size of the auxiliary dataset
>  The auxiliary dataset used in our case study contains 324 individuals. We have now explicitly clarified this in the main text to avoid confusion.
>
> ---
>
> We are grateful for the opportunity to revise our work and believe that these changes have significantly strengthened the manuscript. We hope the revised version addresses the reviewers' concerns and communicates the contribution of our work.

---

> > ### Comment · Reviewer_Jw35 · 2025-08-05
> >
> > I thank the authors for their detailed responses which have helped me to better understand the novelty and contribution of the paper. Could you provide a bit more detail on the additional simulation study with $\alpha$ misspecified, especially the quantitative findings? Similarly, could you quickly summarize what adjustments are required to generalize the relationship between $Y$ and $W$ in the way you describe and what exactly changes in the result? An "arbitrary nonlinear function" seems quite general considering that the authors first assumed linearity.
> >
> > I am a bit concerned about the authors' claims in their rebuttal to Reviewer ScZv that the paper does not "require the errors to follow a normal distribution." Normal distribution of the additive error terms is explicitly required in Equations (1) and (2). In the proof of Theorems 1-4 in Appendix B, this requirement is explicitly used (line 659) and the Gaussian PDF is inserted. Thus, it is unclear to me how the authors' derivations would hold for completely general finite-variance noise.

---

> ### Author Response · Authors · 2025-08-07
> **Simulation Results and Further Discussion**
>
> ### Simulation Study for Misspecified $\alpha$
>
> #### Data Generating Process
> - $X_1 \sim \text{Uniform}[1,2]$
> - $X_2 \sim \text{Uniform}[0,1]$
> - $W = \phi T + X_1 + \text{noise}$
> - $Y = \alpha(X) W + \text{noise}$ where $\alpha(X) = X_1 + a X^2_2$
> - $\alpha_{\text{mis}}(X) = X_1$ (misspecified model)
> - $\zeta \in \{0,1,2\}$
> - $\phi \in \{-1,0,1\}$
>
> | $a$ | $\phi$ | avg. Bias |
> |-----|--------|-----------|
> | 0   | -1     | 0.000985  |
> | 0   | 0      | 0.000127  |
> | 0   | 1      | -0.000731 |
> | 1   | -1     | -0.156842 |
> | 1   | 0      | 0.000127  |
> | 1   | 1      | 0.157096  |
> | 2   | -1     | -0.313669 |
> | 2   | 0      | 0.000127  |
> | 2   | 1      | 0.313923  |
>
> The results demonstrate a clear pattern:
> 1. When $a = 0$ (no misspecification), bias is negligible
> 2. As $a$ increases (greater misspecification), the magnitude of bias increases proportionally for non-zero $\phi$
> 3. When $\phi = 0$, bias remains consistently negligible regardless of $a$
>
> ### Regarding Linearity
>
> Thank you for your question about (non)linearity. We would like to emphasize that the linear case is the most insightful scenario in our analysis. If efficiency gains are unattainable in the linear case, they will necessarily be unattainable in the non-linear case as well.
>
> It's worth noting that when $\zeta$ (the correct basis) is known, even an arbitrary non-linear scenario reduces to the linear case. However, when $\zeta$ is unknown (even if $\alpha$ is known), we return to the fundamental limitation where asymptotic gains are not possible.
>
> ### Gaussian Noise
>
> Thank you for the clarification regarding additive Gaussian noise. In the current version of our paper, we do leverage mean-zero Gaussian properties and homoskedastic noise to derive the efficient influence function. However, this was primarily for parsimony and is not fundamental to our main claims about efficiency gains.
>
> A more general result on the derivation of efficient influence functions (EIF) follows directly from Section 2 of Chernozhukov et al. (2018) "Double/Debiased Machine Learning for Treatment and Causal Parameters," which derives EIFs under mild regularity conditions including log-differentiability and finite variance. The literature on semiparametrically efficient estimation generally requires only additive mean-zero noise with finite non-zero variance, not specifically Gaussian noise.
>
> For clarity, consider the derivation of $R^*_0(O; \theta_0, \eta_0)$. Following equation 2.5 in Chernozhukov et al. (2018), let $\mathcal{L}(O; \theta,\eta)$ be a criterion function maximized when $\theta=\theta_0$ and $\eta=\eta_0$. Under mild regularity conditions, we can use $\mathcal{L}$ to characterize the Neyman orthogonal score function.
>
> Following their Example 2.1, we can derive $R^*_0(O; \theta_0, \eta_0)$ using negative mean squared error (quasi log-likelihood) as our $\mathcal{L}$:
>
> $$\mathcal{L} = -\frac{(Y - \mu_Y(X) - \theta(X)(T-\mu_T(X)))^2}{2}$$
>
> Using the concentrating-out approach from Newey (1994), we obtain:
>
> $$R^*(O; \theta_0, \eta_0) = R_\theta - \Pi(R_\theta | \Lambda_{\eta}) = (Y - \mu_Y(X) - \theta(X)(T-\mu_T(X)))(T-\mu_T(X))$$

---

### Official Review · Reviewer_BAwh · 2025-07-02

**Clarity:** 2
**Significance:** 4
**Originality:** 4
**Rating:** 5
**Confidence:** 3

**Summary:**

Experimental studies often have low population sizes, so it would be beneficial to be use all of the data to draw conclusions about treatment effect.  However, one potential pitfall is that there may be multiple ways to measure a given outcome.  If these measures are not simple transformations of each other (e.g., Fahrenheit vs Celsius), it's less straightforward to use the results from one population to aid in drawing conclusions about the other, and, if the mapping is complex/opaque enough, there may not be any benefit at all.  The authors explore the situations in which such generalization can be done.  They break down three assumption sets based on the level of knowledge about the mapping and give clear descriptions of when using information from a secondary population would be useful.  They present results from a pair of empriical studies to show how their approach can be used in practice.

**Questions:**

It's possible I'm missing something, but I'm not following the terminology in the Influence Function section (the equation after line 145).  In Section 3, you defined $P_n(f(A))$ as the average and $P(f(A))$ as the population average of some function $f$.  However, in this equation, we see $P_n$ and $P$ with no function.  Unless there was some statement that I missed about what they mean standalone, I'm not sure how to interpret $(P_n - P)$ at the beginning of that equation.

As I noted under Strengths and Weaknesses, it seems as though the XBOT and POAT studies have different treatment definitions.  Is this correct?  If so, does that mismatch introduce any issues?

**Ethical Concerns:**

["NO or VERY MINOR ethics concerns only"]

**Final Justification:**

Ultimately, I think this paper is a valuable contribution.  Especially in fields or specific problems with sparser data availability, it makes sense that people would want to combine study results, and I think this paper takes a very cautious and reasonable approach to addressing this.  I especially like the fact that the takeaway is not just "Here's a fancy new method for integrating study results" but rather "You can integrate study results, but only if you're willing to make some pretty strong assumptions that may not hold in practice."  This sort of science - exploring the limits and assumptions around when you /can't/ do something - can be incredibly valuable to the community, and I think this paper is worth publishing for that reason.  There are some clarity concerns from other reviewers that I hope the authors address, but nothing that I feel is fatal.

**Limitations:**

Yes, I believe the authors discussed the limitations sufficiently

**Paper Formatting Concerns:**

No formatting issues

**Quality:**

3

**Strengths And Weaknesses:**

I think the problem that this paper is tackling is an important one, and I think the way the way the authors tackle it (showing the utility as assumptions strengthen) is compelling.  Ultimately, I think this is a good paper, though there are some clarity issues that could be improved.  I would be happy to increase my score if these issues are addressed and my questions are answers.

From Section 3, it sounds like the assumption is that the treatment and covariates are the same between the two studies and that the outcome is the only piece that's allowed to differ in definition/measure.  However, looking at the two studies used for the experimental results, I'm not seeing this assumption being met.  It looks like for the XBOT trial, treatment is XR-NTX vs BUP-NX, while for the POAT study, treatment is BUP-NX vs BUP-NX with counseling.  So while both have BUP-NX as one of the treatment values, the other value seems to be non-trivially different.  They also seem to have different treatment schedules.  It's very possible I'm missing something here, but if not, it's strange to not mention this when it seems to violate an assumption of your approach.

The experimental results are interesting, but I could use another sentence or two of analysis.  From this, would you conclude that the studies show that BUP-NX is likely more effective than XR-NTX?  Does this suggest any future avenues of study, or that additional data should be collected?  It's almost there, but with the conclusion stating that the experiments "[demonstrate] the framework's practical utility", some mentions of the actual practical takeaways would help.

If you're going to list simulation study results as one of your contributions, it would be helpful to at last include a short summary of the takeaways from those experiments in the main paper.  As it is, it's strange that they're not mentioned outside of stating their existence in the introduction and conclusion.

Sections 4.3 and 4.4 seem to be where you do the main summarizing of your results.  However, I think a higher level summary before going into Section 5, making it very clear what the key takeaways are and what the conditions are that allow for any gains from using the second population, would aid with clarity.

---

> ### Author Rebuttal · Authors · 2025-07-29
>
> We sincerely thank the reviewer for their thoughtful and constructive comments. Below, we address the key concerns raised and summarize the corresponding updates made to the manuscript:
>
> **1. Clarifying the Use of Auxiliary Data Despite the Absence of XT-NTX**
>
> Thank you for this thoughtful and nuanced observation. You are absolutely right—the XT-NTX arm is not present in the auxiliary study. However, since BUP-NX is common to both the primary and auxiliary datasets, our goal is to assess whether and under what conditions information on BUP-NX from the auxiliary study can be leveraged to improve the efficiency of estimating $\mathbb{E}[Y(t=\text{BUP-NX}) \mid S=0]$ in the primary study population. We have revised the manuscript to clarify this point. The updated text reads:
>
> > *“Although the XT-NTX arm is absent in the auxiliary dataset, the BUP-NX treatment is shared across both studies. We aim to evaluate if and when information on outcome response to BUP-NX from auxiliary study can be used to improve the efficiency of estimating the outcome under BUP-NX in the primary population, $\mathbb{E}[Y(t=\text{BUP-NX}) \mid S=0]$, while carefully considering the assumptions required for valid data fusion.”*
>
> **2. Main Takeaways from the Case Study**
>
> Thank you so much. We have added the following to describe the main takeaways of our case study:
>
> > *“In our case study, we assess whether XT-NTX is more effective than BUP-NX in reducing withdrawal symptom severity. The point estimate from the primary study is slightly negative, suggesting a marginal advantage for XT-NTX, but the 95% confidence interval includes zero, indicating no statistically significant difference between the two treatments (see Figure 1(b); estimate $\hat{\theta}_0$). To improve estimation precision, we explore leveraging auxiliary data. Under the strong assumption, our combined analysis yields a statistically significant result favoring BUP-NX over XT-NTX (see Figure 1(b); estimate $\hat{\theta}_a$). While such findings may appear actionable, they hinge critically on an untestable assumption linking outcomes $Y$ and $W$. If this assumption is violated, the resulting estimates may be misleadingly precise. Our case study thus serves as a cautionary example: although integrating auxiliary data can improve precision, it must be done with careful scrutiny of the underlying assumptions, which—if invalid—can lead to confidently incorrect conclusions.”*
>
> **3. Placement of the Simulation Study**
>
> Thank you so much for your helpful suggestion. Due to space constraints, we initially moved the simulation studies to the appendix. However, following your recommendation—and now having an extra page available for the camera-ready version—we have moved the simulation study back into the main text. We believe this change strengthens the paper by making the empirical insights more accessible and better integrated with the theoretical results.
>
> **4. Summary of Theoretical Takeaways**
>
> Thank you so much for the extremely constructive input. We have added **Subsection 4.5**, which summarizes the main theoretical and technical takeaways from our results.
>
> **5. Clarifying Notation: $(P_n - P)(\psi)$**
>
> Thank you for the question. The expression $(P_n - P)(\psi)$ denotes the difference between the empirical average $P_n(\psi)$ and the population expectation $P(\psi)$ of the influence function $\psi$. This is a standard shorthand in semiparametric statistics and causal inference literature, used frequently to express asymptotic linearity and derive variance for regular estimators (see, for example, Kennedy 2016; Tsiatis 2006). To avoid confusion, we have clarified the notation in the revised manuscript. The updated text reads:
>
> > *“Here, $P_n(\psi)$ denotes the empirical average and $P(\psi)$ the population expectation. The notation $(P_n - P)(\psi)$ is shorthand for $P_n(\psi) - P(\psi)$, as commonly used in semiparametric theory.”*
>
> **6. Necessity of Shared Treatment**
>
> Thank you for raising this important point. The goal of our paper is to understand whether, and under what assumptions, auxiliary data can be used to improve efficiency when the outcomes differ across studies. In this context, at least one treatment arm must be shared between the primary and auxiliary studies. Without a common treatment and in the presence of disparate outcomes, it becomes fundamentally impossible to leverage auxiliary data for valid causal inference. We have updated the discussion section to clarify this point. The revised text reads:
>
> > *“Our framework requires that at least one treatment arm be shared across the primary and auxiliary studies. In the absence of such overlap, and when outcomes differ across studies, there is no basis for linking the potential outcome distributions across datasets, rendering auxiliary data less usable for improving efficiency gains. This highlights a key structural limitation in data fusion settings with disparate outcomes.”*

---

> > ### Comment · Reviewer_BAwh · 2025-08-06
> >
> > Thank you for your detailed response!
> >
> > The clarification that the treatments don't need to exactly match but rather share at least one treatment arm is significant.  I think the added text you've proposed is good, but it also sounds like you're only proposing adding such clarification to the experimental setup and discussion sections, when it really feels like it should be at least brought up earlier.  In Section 3, you state "In both studies, we observe treatment T $\in$ {0,1} and covariates X", which really seems to imply that the treatment definitions match between both studies.  Since it sounds like that's not the case (the OS could have T $\in$ {0,1} while the RCT has T $in$ {0,2}), it would help to bring that up here so the reader isn't surprised when that's suddenly changed in the experimental results section.
> >
> > In addition, you state that at least one treatment arm needs to be shared between both studies.  In the POATS and XBOT studies, while both do have a BUP-NX arm, the way that treatment is administered (the treatment and follow-up schedule) differs between them.  I don't know much at all about this domain, so it's possible that these differences are relatively minor in practice, and I don't necessarily think it's a bad thing that they're not an exact match, since, if anything, it makes it a more realistic case study.  But I think just stating that a treatment arm needs to match suggests that it needs to be an exact match.  The fact that, in practice, non-exact matches may be close enough is actually pretty cool and I think warrants at least a sentence or two in the discussion.
> >
> > Ultimately, I think this is a cool paper.  It has a bit of a 'negative results' aspect to it (hence the "Cautionary Tale" title), which can often make papers hard to publish even when such "negative results" (in this case, the fact that you often can't gain useful information from other studies unless you're willing to make some strong assumptions) are often extremely valuable and informative to the community.  The added clarification around the data match requirements and the inclusion of the simulation studies in the main paper are great changes, and I think the extra space provided for the camera ready will allow the authors to make the paper a clearer read overall.  I'm happy to increase my score from a 4 to a 5.

---

### Decision · Program_Chairs · 2025-09-17

**Decision:**

Accept (poster)

**Comment:**

My recommendation is to accept the paper. This decision was close because there was strong disagreement among reviewers. I hope that the authors will take all feedback into account when making revisions for the camera-ready.

Three reviewers found the message to be a valuable contribution, warning against a plausible pitfall in data integration. There were some issues raised by these reviewers, but overall they supported the message, the analysis, and the example given in the main text.

One reviewer found the paper under-motivated. To address this concern, I think it would be useful for the authors to include reference(s) to at least one study that attempts this data integration in the current literature (perhaps as part of a meta-analysis?). This would motivate the need for a cautionary note against current or emerging practice.

It would also be useful to clarify the scope of the theory. In specific, being very clear that the theoretical claims are about ATE estimation, and not CATE estimation would be useful. More broadly, explicitly stating the scope of the conclusions from the theory (i.e., that if things don't work in the restrictive linear context, we would expect them to fail in more complex contexts) would be useful.

Better integrating the simulated demonstrations of the theory into the main text, as some reviewers requested, would also be useful.